# Generalizing Consistent Multi-Class Classification with Rejection to be Compatible with Arbitrary Losses

**Yuzhou Cao**[1*]    **Tianchi Cai**[2]    **Lei Feng**[3,4†]    **Lihong Gu**[2]
**Jinjie Gu**[2]    **Bo An**[1]    **Gang Niu**[4]    **Masashi Sugiyama**[4,5]
[1]School of Computer Science and Engineering, Nanyang Technological University, Singapore
[2]Ant Group, China
[3]College of Computer Science, Chongqing University, China
[4]RIKEN Center for Advanced Intelligence Project, Japan
[5]The University of Tokyo, Japan

## Abstract

*Classification with rejection* (CwR) refrains from making a prediction to avoid critical misclassification when encountering test samples that are difficult to classify. Though previous methods for CwR have been provided with theoretical guarantees, they are only compatible with certain loss functions, making them not flexible enough when the loss needs to be changed with the dataset in practice. In this paper, we derive a novel formulation for CwR that can be equipped with arbitrary loss functions while maintaining the theoretical guarantees. First, we show that $K$-class CwR is equivalent to a $(K+1)$-class classification problem on the original data distribution with an augmented class, and propose an empirical risk minimization formulation to solve this problem with an estimation error bound. Then, we find necessary and sufficient conditions for the learning *consistency* of the surrogates constructed on our proposed formulation equipped with any classification-calibrated multi-class losses, where consistency means the surrogate risk minimization implies the target risk minimization for CwR. Finally, experimental results validate the effectiveness of our proposed method.

## 1 Introduction

In risk-sensitive multi-class classification applications (e.g., medical diagnosis, healthcare, autonomous driving, and product inspections [13, 22, 44]), misclassification can cause serious or even fatal consequences. To alleviate this issue, many studies have been conducted on *classification with rejection* (CwR) [11, 6, 63, 13, 14, 16, 22, 52, 48, 44, 8], which can abstain from making an unsure prediction to prevent such critical misclassification.

Most of the previous studies follow the framework that provides the reject option with a pre-defined cost $c$ which is lower than the misclassification cost 1. Given cost $c$, the problem is further formulated as a risk minimization problem that aims to minimize the expectation of the zero-one-$c$ loss, i.e., the zero-one-$c$ risk. With the risk minimization process, the obtained classifier can balance the cost of rejection and prediction by choosing to incur a rejection cost $c$ if the misclassification risk is high.

Due to the discontinuous nature of the zero-one-$c$ loss, recent works focused on finding its continuous surrogates to make the optimization problem tractable. A basic requirement for surrogate losses is the *consistency* [65, 7, 54, 47], i.e., the surrogate risk minimization implies the zero-one-$c$ risk minimization. Moreover, compared with the traditional $K$-class classification task where decisions

---

*Work done when Yuzhou Cao was a research intern at Ant Group.
†Corresponding author: Lei Feng <lfeng@cqu.edu.cn>

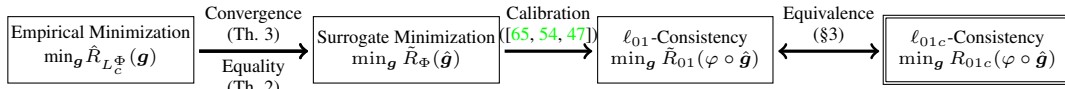

**Figure 1:** Overview of the construction of consistent surrogates for classification with rejection in this work.

are normally made from the index of the maximum coordinate of a $K$-dimensional scoring function, the design of decision criteria in the CwR task is more elusive due to the existence of a reject option. By adopting different classification and rejection criteria, various surrogates of the zero-one-$c$ loss have been proposed with consistency analyses [6, 63, 13, 14, 48, 44, 8].

Classical studies focused on developing *confidence-based methods*[6, 63, 48, 44], which use the outputs of classifiers as confidence values and set a real-valued threshold as the rejection rule. Representative methods [63, 44] used surrogates that depend on *class-posterior probability estimation* (CPE) [50, 58], which is challenging when using deep models [25]. Though some of them [6, 48, 35, 24] could avoid CPE, most of them applied the modification of non-differentiable hinge/ramp-like surrogates, and their performance was only validated with linear models.

To avoid the use of the confidence threshold, Cortes et al. [13] provided an upper bound of the zero-one-$c$ loss as the surrogate that allows the use of a separated rejector and can be trained simultaneously with the classifier, which is regarded as *classifier-rejector methods*. Though these methods achieved state-of-the-art performance in binary classification scenarios, they only provided a consistency guarantee for hinge-like and exponential losses and cannot be directly generalized to the multi-class scenario as shown in Ni et al. [44]. Charoenphakdee et al. [8] showed that $K$-class CwR can be decomposed into $K$ binary cost-sensitive classification problems [17, 51, 12] and proposed a family of surrogates are the ensembles of arbitrary binary classification losses, which can avoid CPE and the use of confidence threshold with properly chosen losses when the cost function is constant. Mozannar and Sontag [41] provided a modified version of the cross entropy loss as the surrogate for the task of learning to defer [41, 42] that can also be used in CwR, while its optimal solution still relies on CPE. In summary, previous works only took limited types of losses into consideration, and there lacks a theoretically grounded framework that can cover all the surrogates used in multi-class classification.

In this paper, we propose a novel framework for CwR that allows the use of arbitrary surrogate losses used in traditional multi-class classification as long as they are classification-calibrated, including but not limited to the well-known cross entropy loss, mean absolute error, focal loss [33, 9], and the pairwise/one-versus-all generalizations of binary margin losses [65]. Thanks to the flexible choices of losses, we are free of the restricted analyses on the consistency of certain surrogates. The access to a full range of surrogates also make it possible to custom the loss contingent on the characteristics and actual requirements of different datasets and tasks. An overview of our framework is shown in Figure 1. We summarize the main contributions of this work as follows:

- We disclose the equivalence between $K$-class CwR and a $(K+1)$-class classification problem on the original data distribution with an augmented class, by showing the equality between their classification risks.

- We propose a formulation of surrogates for $\ell_{01c}$ that can recover the surrogate risk of a $(K+1)$-class classification task only with the $K$-class training distribution, and derive an estimation error bound for its empirical risk minimization.

- We find a sufficient condition for the consistency of the proposed family of surrogates *w.r.t.* the zero-one-$c$ loss that allows the use of any calibrated multi-class surrogates and show its necessity when considering a large family of loss functions.

- We for the first time provide an analysis on the calibration of the *generalized cross entropy* loss [66] that benefits from both the cross entropy loss and mean absolute error, and experimentally demonstrate that it is suitable for our proposed framework.

## 2 Preliminaries

In this section, we provide preliminary knowledge of CwR and calibrated surrogate losses, and discuss the consistency in CwR.

## 2.1 Classification with Rejection

The problem setting of CwR is based on the cost-based framework [11]. Let us denote by $\mathcal{X}$ the feature space, $\mathcal{Y} = \{1, 2, \ldots, K\}$ the label space , and $\mathcal{Y}^{\circledR} = \{1, 2, \ldots, K, \circledR\}$ the label space with a reject option. We are given instance-label pairs $\{(\boldsymbol{x}_i, y_i)\}_{i=1}^n$ independently and identically drawn from an underlying distribution with probability density $p(\boldsymbol{x}, y)$. The goal of CwR is to train a classifier $f : \mathcal{X} \to \mathcal{Y}^{\circledR}$ that can abstain from making a decision, where $\circledR$ denotes the reject option. The evaluation metric of this task is the zero-one-$c$ loss $\ell_{01c}$, which can be expressed as a variant of the traditional zero-one loss $\ell_{01}(f(\boldsymbol{x}), y) = [\![f(\boldsymbol{x}) \neq y]\!]$:

$$\ell_{01c}(f(\boldsymbol{x}), y) = \begin{cases} c, & f(\boldsymbol{x}) = \circledR, \\ [\![f(\boldsymbol{x}) \neq y]\!], & f(\boldsymbol{x}) \in \{1, 2, \ldots, k\}, \end{cases}$$

where $[\![\cdot]\!]$ is the Iverson bracket notation as suggested by Knuth [29] and the cost $c$ can be further extended to an instance-dependent function $c(\boldsymbol{x})$. Our goal is to train a classifier that can minimize the expectation of $\ell_{01c}$ over the data distribution:

$$R_{01c}(f) = \mathbb{E}_{p(\boldsymbol{x}, y)}[\ell_{01c}(f(\boldsymbol{x}), y)]. \tag{1}$$

Let us denote by $f^* = \operatorname{argmin}_f R_{01c}(f)$ the Bayes optimal solution and $\boldsymbol{\eta}(\boldsymbol{x}) = \{p(y|\boldsymbol{x})\}_{y=1}^K$ the posterior probabilities. When evaluated by $\ell_{01c}$, a classifier receives a standard classification error in $\{0, 1\}$ if it makes a prediction and a cost of $c$ if it does not make a prediction (i.e., chooses the reject option). Intuitively, an optimal solution $f^*$ should balance the possibility of misclassification and the rejection cost $c$. This explanation is theoretically justified by Chow's rule [11]:

**Definition 1.** (Chow's Rule) A classifier $f : \mathcal{X} \to \mathcal{Y}^{\circledR}$ is the optimal solution of (1) if and only if it meets the following condition almost surely:

$$f(\boldsymbol{x}) = \begin{cases} \circledR, & \max_y \eta_y(\boldsymbol{x}) \leq 1 - c, \\ \operatorname{argmax}_y \eta_y(\boldsymbol{x}), & \text{else.} \end{cases}$$

Chow's rule shows that the optimal solution should refrain from making a decision if the most confident prediction of an example is still not confident enough given a rejection cost $c$.

## 2.2 Calibrated Surrogate Losses

Most classification problems can be formalized as the minimization of the target risk, which is the expectation of a target loss. Then *empirical risk minimization* (ERM) is conducted to obtain models with performance guarantees. However, most of the target losses are discontinuous, e.g., the zero-one loss in multi-class classification and the Hamming/ranking loss in multi-label classification [20]. Therefore, directly optimizing them is usually difficult and even NP-hard [18].

In order to optimize the target risk efficiently, surrogate risk minimization is preferred that minimizing the expectation of a continuous surrogate loss instead, e.g., the hinge loss in binary classification and the cross entropy loss in multi-class classification. For the statistical consistency of learning, *calibration* [53] is considered as a basic requirement for surrogate losses, which is a pointwise version of consistency and means that the minimization of the surrogate loss yields that of the target loss for each possible sample. A commonly adopted definition of the calibration of surrogates in multi-class classification is given as follows:

**Definition 2.** ($\ell_{01}$-Calibration [7, 54, 47]) For a $K$-class classification problem with target loss $\ell_{01}$, we say $\Phi : \mathbb{R}^K \times \mathcal{Y} \to \mathbb{R}_+$ is $\ell_{01}$-calibrated if for any $\boldsymbol{p} \in \Delta^K$:

$$\inf_{\boldsymbol{u} \in \mathbb{R}^K, \boldsymbol{u} \notin \operatorname{argmin}_{\boldsymbol{u}} \boldsymbol{p}^T \boldsymbol{L}_{01}(\boldsymbol{u})} \boldsymbol{p}^T \boldsymbol{\Phi}(\boldsymbol{u}) > \inf_{\boldsymbol{u} \in \mathbb{R}^K} \boldsymbol{p}^T \boldsymbol{\Phi}(\boldsymbol{u}),$$

where $\boldsymbol{\Phi}(\boldsymbol{u}) = \{\Phi(\boldsymbol{u}, y)\}_{y=1}^K$, $\boldsymbol{L}_{01}(\boldsymbol{u}) = \{\ell_{01}(\operatorname{argmax}_{y' \in \mathcal{Y}} \boldsymbol{u}_{y'}, y)\}_{y=1}^K$.

The definition of $\ell_{01}$-calibration requires that a surrogate loss should be able to distinguish between optimal solutions and non-optimal ones *w.r.t.* any potential posterior distribution $\boldsymbol{p}$. This property is shown to be a necessary and sufficient condition for the statistical consistency of surrogate risk minimization, and fruitful research on the verification of $\ell_{01}$-calibrated surrogates has been conducted [7, 65, 54, 47, 46, 19].

**Table 1:** Comparisons between our proposed method and previous works of multi-class classification with rejection. Since our method is induced from a $(K+1)$-class classification problem, we can render a consistent learning guarantee with arbitrary surrogate losses that are calibrated *w.r.t.* the zero-one loss. Thanks to the abundant choices of losses, our proposed method can avoid CPE and the use of confidence thresholds.

| Method | CPE-Free | Instance-Dependent Cost | Confidence Threshold-Free | Arbitrary Losses |
|:---:|:---:|:---:|:---:|:---:|
| [48] | ✓ | ✓ | ✗ | ✗ |
| [44] | ✗ | ✓ | ✗ | ✗ |
| [41] | ✗ | ✓ | ✓ | ✗ |
| [8] | ✓ | ✗ | ✓ | ✗ |
| Proposed | ✓ | ✓ | ✓ | ✓ |

Besides multi-class classification, the calibration of surrogate losses also has been studied in various aspects of statistical learning, including but not limited to, multi-label classification [20, 64, 30, 59], AUC optimization [21, 39], general linear-fractional utility maximization [3], cost-sensitive learning [12, 51], top-$K$ classification [32, 62], and adversarially robust classification [4, 2, 1].

## 2.3 Consistency in Classification with Rejection

' In the field of CwR, we are also interested in the consistency of surrogate losses. Let $\mathcal{C} \subset \mathbb{R}^d$ where $d \in \mathbb{N}$ and $\Phi : \mathcal{C} \times \mathcal{Y} \to \mathbb{R}_+$ is a surrogate loss, the consistency is defined as follows:

**Definition 3.** ($\ell_{01c}$-Consistency) A surrogate loss $\Phi : \mathcal{C} \times \mathcal{Y} \to \mathbb{R}_+$ is $\ell_{01c}$-consistent if there exists a function $\varphi : \mathcal{C} \to \mathcal{Y}^\circledR$ for all probability distributions and all the sequences of functions $\{\boldsymbol{g}_i\}_{i\in\mathbb{N}} : \mathcal{X} \to \mathcal{C}$:

$$R_\Phi(\boldsymbol{g}_i) \to R_\Phi^* \Rightarrow R_{01c}(\varphi \circ \boldsymbol{g}_i) \to R_{01c}^*, \tag{2}$$

where $R_\phi(\boldsymbol{g}) = \mathbb{E}_{p(\boldsymbol{x},y)}[\Phi(\boldsymbol{g}(\boldsymbol{x}), y)]$, $R_\Phi^* = \inf_{\boldsymbol{g}:\mathcal{X}\to\mathcal{C}} R_\Phi(\boldsymbol{g})$, and $R_{01c}^* = \inf_{f:\mathcal{X}\to\mathcal{Y}^\circledR} R_{01c}(f)$.

This definition is inspired by the problem of general multi-class classification [47]. For an $\ell_{01c}$-consistent surrogate loss $\Phi$, we can safely minimize the surrogate risk $R_\Phi$ instead while retaining the consistency guarantee of $R_{01c}$.

To ensure the consistency of $\Phi$, it is routine to discuss the calibration of surrogate losses. However, unlike the classical multi-class classification problem, where $\varphi$ is usually an argmax operator, the design of $\varphi$ for CwR can be quite complicated and hard to be unified, which makes it difficult to directly do calibration analysis on $\Phi$. The flexibility of $\varphi$ also limits the discussions to specific types of surrogate losses. In Ramaswamy et al. [48], the authors considered the multi-class extensions of the hinge-loss with a confidence threshold. Ni et al. [44] indicated that the confidence-based method is indispensable and only focuses on class probability estimation via surrogate risk minimization. Both of Mozannar and Sontag [41] and Charoenphakdee et al. [8] gave surrogate losses for the zero-one-$c$ loss that does not depend on the accurate estimation of the class probability, while Mozannar and Sontag [41] focused on a variant of the cross entropy loss and Charoenphakdee et al. [8] constructed calibrated surrogate losses with the ensemble of $K$ calibrated losses for binary classification.

In this paper, instead of directly discussing the calibration of surrogate $\Phi$, we show that there is an equivalence between classical multi-class classification and CwR. Based on this equivalence, we show that it is sufficient for $\Phi$ to be $\ell_{01c}$-consistent by letting it be a simple variant of **any** calibrated surrogate loss *w.r.t.* the traditional zero-one loss $\ell_{01}$. The comparison of the proposed method and related works is shown in Table 1.

## 3 Equivalence between Classification with Rejection and Ordinary Classification

In this section, we first show that the risk $R_{01c}(f)$ can be formalized as a $(K+1)$-class classification problem, and show that we can obtain $\ell_{01c}$-consistent surrogates with a variant of any calibrated surrogate *w.r.t.* $\ell_{01}$, which enables the use of $\mathcal{C} \subset \mathbb{R}^{K+1}$ and $\varphi(\cdot) = \operatorname{argmax}(\cdot)$ as in the traditional multi-class classification tasks. We also show that such equivalence also holds when the cost $c$

depends on sample $\boldsymbol{x}$. The proof of the conclusions in this section can be found in Appendix A. We start by considering the following distribution $\mathcal{D}_c^\circledR$ over $\mathcal{X} \times \mathcal{Y}^\circledR$ with probability density $\tilde{p}(\boldsymbol{x}, \tilde{y})$:

**Definition 4.** (Self-Augmented Distribution) A distribution $\mathcal{D}_c^\circledR$ is called a $c$-self-augmented distribution $w.r.t.$ $\mathcal{D}$ if its probability density meets the following conditions:

$$\tilde{p}(\boldsymbol{x}, \tilde{y}) = \begin{cases} \frac{p(\boldsymbol{x}, y)}{2-c}, & \tilde{y} \in \{1, 2, \ldots, K\}, \\ \frac{(1-c)p(\boldsymbol{x})}{2-c}, & \tilde{y} = \circledR. \end{cases}$$

It can be seen that distribution $\mathcal{D}_c^\circledR$ shares the same marginal density of $\boldsymbol{x}$ as the original distribution $\mathcal{D}$ while $\mathcal{D}_c^\circledR$ has an augmented class $\circledR$ with class probability determined by the rejection cost $c$. Based on the connection between $\mathcal{D}_c^\circledR$ and $\mathcal{D}$, we can further explore the relation between the two tasks: classification on $\mathcal{D}_c^\circledR$ and CwR on $\mathcal{D}$.

**Theorem 1.** For any classifier $f : \mathcal{X} \to \mathcal{Y}^\circledR$, the following equation holds:

$$R_{01c}(f) - R_{01c}^* = (2 - c)\left(\tilde{R}_{01}(f) - \tilde{R}_{01}^*\right),$$

where $\tilde{R}_{01}(f) = \mathbb{E}_{\tilde{p}(\boldsymbol{x}, \tilde{y})}[\ell_{01}(f(\boldsymbol{x}), \tilde{y})]$ and $\tilde{R}_{01}^* = \inf_{f : \mathcal{X} \to \mathcal{Y}^\circledR} \tilde{R}_{01}(f)$.

Theorem 1 reveals the equivalence between the two tasks in a straightforward manner. Since the multiplication of the classification risk on $\mathcal{D}_c^\circledR$ with a positive constant is equal to $R_{01c}(f)$, the minimization of $\tilde{R}_{01}(f)$ immediately yields the minimization of $R_{01c}(f)$ and vice versa. Furthermore, according to the linear correlation between $\tilde{R}_{01}(f)$ and $R_{01c}(f)$, we can directly quantify the excess error $R_{01c}(f) - R_{01c}^*$ by bounding $\tilde{R}_{01}(f) - \tilde{R}_{01}^c$, which is an easier work thanks to the existing research of multi-class classification. In conclusion, risk minimization with $\tilde{R}_{01}(f)$ can also give a classifier with a rejection option with the optimality guarantee, and then we can consider a surrogate risk minimization problem for multi-class classification instead of CwR.

When the cost $c(\boldsymbol{x})$ is an instance-dependent function, we show that such equivalence still holds with a minor modification. Considering the reweighted zero-one loss: $\bar{\ell}_{01}(f(\boldsymbol{x}), y) = (2-c(\boldsymbol{x}))[\![f(\boldsymbol{x}) \neq y]\!]$ and its expectation $\bar{R}_{01}(f)$ on $\mathcal{D}_c^\circledR$, we have the following conclusion:

**Corollary 1.** For any classifier $f : \mathcal{X} \to \mathcal{Y}^\circledR$, the following equation holds:

$$\bar{R}_{01}(f) - \bar{R}_{01}^* = R_{01c}(f) - R_{01c}^*.$$

It is obvious that Theorem 1 is a special case of Corollary 1 with constant cost functions. Though here we consider a reweighted classification task, the calibration result of multi-class surrogate losses can still be applied without any modification since the minimization of $\bar{R}_{01}(f)$ can be seen as ordinary classification risk minimization with a slightly different marginal density $p'(\boldsymbol{x})$, which does not affect the calibration result since the class-posterior possibilities remain unchanged. All the conclusions in the rest of this paper can be extended to the scenario of instance-dependent cost (see Appendix G).

## 4 $\ell_{01c}$-Consistent Surrogates with Arbitrary $\ell_{01}$-Calibrated Losses

As discussed in Section 3, CwR can be safely replaced by multi-class classification on a special distribution $\mathcal{D}_c^\circledR$. Following the surrogate risk minimization in multi-class classification, we can replace the zero-one loss $\ell_{01}$ with a surrogate risk $\Phi : \mathbb{R}^{K+1} \times \mathcal{Y} \cup \{K+1\} \to \mathbb{R}_+$ and minimizing the surrogate risk with a score-based classifier $\boldsymbol{g} : \mathcal{X} \to \mathbb{R}^{K+1}$ instead, which is defined as follows:

$$\tilde{R}_\Phi(\boldsymbol{g}) = \mathbb{E}_{\tilde{p}(\boldsymbol{x}, \tilde{y})}[\Phi(\boldsymbol{g}(\boldsymbol{x}), t(\tilde{y}))], \tag{3}$$

where $t(\tilde{y}) = K + 1$ if $\tilde{y} = \circledR$ and $t(\tilde{y}) = \tilde{y}$ otherwise. (3) is a typical formulation of the multi-class classification risk and we can asymptotically minimize it following the ERM framework [55]. After the risk minimization process, the prediction is generated with the following link function $\varphi : \mathbb{R}^{K+1} \to \mathcal{Y}^\circledR$:

$$\varphi(\boldsymbol{u}) = \begin{cases} \circledR, & \arg\max\limits_{y \in \mathcal{Y} \cup \{K+1\}} \boldsymbol{u}_y(\boldsymbol{x}) = K + 1, \\ \arg\max\limits_{y \in \mathcal{Y} \cup \{K+1\}} \boldsymbol{u}_y(\boldsymbol{x}), & \text{else.} \end{cases}$$

With a classification-calibrated surrogate $\Phi$, it is sufficient to say that the minimization of $\tilde{R}_\Phi(\boldsymbol{g})$ can lead to that of $\tilde{R}_{01}(\varphi(\boldsymbol{g}))$, which indicates the minimization of $R_{01c}(\varphi(\boldsymbol{g}))$ according to Theorem 1 and Corollary 1. The theory of how to find such surrogates has been thoroughly studied in the field of the classification-calibration of multi-class surrogates [65, 54, 47].

However, we do not have direct access toward $\mathcal{D}_c^\circledR$ though it is closely related to the available data distribution $\mathcal{D}$. In this section, we propose a family of surrogate losses based on the conclusions in the previous section, which allows the use of any multi-class classification surrogates. With this formulation of surrogates, we can recover the classification risk of $\tilde{R}_\Phi(\boldsymbol{g})$ without access to $\mathcal{D}_c^\circledR$ by taking its expectation in $\mathcal{D}$. Based on the loss formulation, we also provide the estimation error bound to show the validity of ERM.

## 4.1 Formulation of Surrogates

Here, we begin with the definition of a family of surrogates for the zero-one-$c$ loss, and then show how it can relate $\mathcal{D}$ and $\mathcal{D}_c^\circledR$. With any multi-class classification loss $\Phi$, we have the following formulation of surrogate losses for $\ell_{01c}$:

**Definition 5.** Given a pre-defined rejection cost $c$, we have the following formulation of surrogate $L_c^\Phi : \mathbb{R}^{K+1} \times \mathcal{Y} \to \mathbb{R}_+$ for CwR:

$$L_c^\Phi(\boldsymbol{u}, y) = \Phi(\boldsymbol{u}, y) + (1 - c)\Phi(\boldsymbol{u}, K + 1), \tag{4}$$

where $\Phi : \mathbb{R}^{K+1} \times \mathcal{Y} \cup \{K + 1\} \to \mathbb{R}_+$ and $\boldsymbol{u} \in \mathbb{R}^{K+1}$.

The proposed surrogate loss is the linear combination of a $(K+1)$ dimensional multi-class classification loss with coefficient determined by the predefined cost $c$. It can also be learned from Appendix A.2 of Charoenphakdee et al. [8] that when $\Phi$ is the softmax cross entropy loss, (4) is equivalent to Mozannar and Sontag [41]. The following theorem reveals the connection between $\tilde{R}_\Phi(\boldsymbol{g})$ and the expectation of $L_c^\Phi$ on $\mathcal{D}$:

**Theorem 2.** For any $\boldsymbol{g} : \mathcal{X} \to \mathbb{R}^{K+1}$ and $R_{L_c^\Phi}(\boldsymbol{g}) = \mathbb{E}_{p(\boldsymbol{x}, y)}[L_c^\Phi(\boldsymbol{g}(\boldsymbol{x}), y)]$:

$$R_{L_c^\Phi}(\boldsymbol{g}) = (2 - c)\tilde{R}_\Phi(\boldsymbol{g}).$$

The proof is provided in Appendix B. From Theorem 2, we can obtain the risk $\tilde{R}_\Phi(\boldsymbol{g})$ without access to $\mathcal{D}_c^\circledR$ with the use of the proposed surrogate (4). Following the common practice, we can finally conduct ERM [55] that minimizes the unbiased estimator of $R_{L_c^\Phi}(\boldsymbol{g})$, which is also that of $\tilde{R}_\Phi(\boldsymbol{g})$ according to Theorem 2:

$$\hat{R}_{L_c^\Phi}(\boldsymbol{g}) = \frac{1}{n} \sum_{i=1}^n L_c^\Phi(\boldsymbol{g}(\boldsymbol{x}_i), y_i) \tag{5}$$

After minimizing $\hat{R}_{L_c^\Phi}(\boldsymbol{g})$ and obtaining the empirically optimal $\hat{\boldsymbol{g}}$, we can use it for predicting with the link function $\varphi \circ \hat{\boldsymbol{g}}$, where $\varphi \circ \hat{\boldsymbol{g}}(\boldsymbol{x}) = \varphi(\hat{\boldsymbol{g}}(\boldsymbol{x}))$.

According to the unbiasedness of (3), it is promising that the induced prediction rule $\varphi \circ \hat{\boldsymbol{g}}$ can approximate Chow's rule (Definition 1). To quantify such approximation, there remains two questions: what is the relation between the minimization of empirical risk $\hat{R}_{L_c^\Phi}(\boldsymbol{g})$ and $R_{L_c^\Phi}(\boldsymbol{g})$, and whether the minimization of $R_{L_c^\Phi}(\boldsymbol{g})$ yields that of $R_{01c}(\varphi \circ \boldsymbol{g})$. We will answer the two problems in Section 4.2 and Section 5, respectively.

## 4.2 Estimation Error Bound

In Section 4.1, we proposed a family of surrogates that can recover the surrogate risk on $\mathcal{D}_c^\circledR$ with only $\mathcal{D}$ and provided an ERM framework to learn the empirically optimal $\hat{\boldsymbol{g}} = \min_{\boldsymbol{g} \in \mathcal{G}} \hat{R}_{L_c^\Phi}(\boldsymbol{g})$. Here we further justify the use of ERM by showing that the minimization of $\hat{R}_{L_c^\Phi}$ can also result in that of $R_{L_c^\Phi}$ with the following estimation error bound.

**Theorem 3.** For any $\delta \in (0, 1)$, suppose the model class of $g_y$ is $\mathcal{G}_y$ and $\boldsymbol{g} \in \mathcal{G}$, where $\mathcal{G}_y \subset \mathcal{X} \to \mathbb{R}$ and $\mathcal{G} \subset \mathcal{X} \to \mathbb{R}^{K+1}$ is composed of $\{\mathcal{G}_y\}_{y=1}^{K+1}$. $\Phi(\cdot, y)$ is $\rho$-Lipschitz continuous and is bounded

by $C_\Phi > 0$. Assume that the identifiable condition holds, i.e., $\min_{\boldsymbol{g} \in \mathcal{G}} R_{L_c^\Phi}(\boldsymbol{g}) = R_{L_c^\Phi}^*$, then the following inequality holds with probability at least $1 - \delta$:

$$R_{L_c^\Phi}(\hat{\boldsymbol{g}}) - R_{L_c^\Phi}^* \leq 4\sqrt{2}(2-c)\rho \sum_{y=1}^{K+1} \mathfrak{R}_n(\mathcal{G}_y) + (2-c)C_\Phi \sqrt{\frac{2\log 2/\delta}{n}}, \qquad (6)$$

where $\mathfrak{R}_n(\mathcal{G}_y)$ is the Rademacher complexity [5] *w.r.t.* $\mathcal{G}_y$ on the distribution with density $p(\boldsymbol{x})$ that often decays in the rate of $O(\frac{1}{\sqrt{n}})$.

We prove this conclusion in Appendix C. From the theorem above, we can learn that with the identifiable condition which is a common assumption with the use of complex models [4, 27, 34], $R_{L_c^\Phi}(\hat{\boldsymbol{g}})$ converges to $R_{L_c^\Phi}^*$ in $O_p(1/\sqrt{n})$, which is the optimal parametric convergence rate without additional assumptions [38]. According to Theorem 2, it is straightforward that $\tilde{R}_\Phi(\boldsymbol{g}) \xrightarrow{P} \tilde{R}_\Phi^*$ also holds. Nevertheless, the relation between the minimization of surrogate risk $\tilde{R}_\Phi(\boldsymbol{g})$ and that of the target risk $\tilde{R}_{01}(\varphi \circ \boldsymbol{g})$ is still unknown. According to Theorem 1, the minimization of $\tilde{R}_{01}(\varphi \circ \boldsymbol{g})$ is equivalent to zero-one-$c$ risk minimization, which is the goal of CwR. We answer this question in the next section by giving a sufficient condition for the $\ell_{01c}$-consistency for $L_c^\Phi$.

# 5 Theoretical Analysis

In this section, we first point out the sufficient condition for $L_c^\Phi$ to be $\ell_{01c}$-calibrated and show that it is also necessary if the candidates of $\Phi$ are permutation-invariant, which consists of a large number of commonly used loss functions, including but not limited to cross-entropy loss, focal loss, and mean absolute error. Then we further specify the regret transfer bounds for a family of CPE-free surrogates [66], which has not been provided with theoretical analysis before.

## 5.1 Necessary and Sufficient Conditions for $\ell_{01c}$-Consistency

Given the loss formulation (4), a natural idea is to construct surrogate $L_c^\Phi$ with commonly used multi-class loss functions. Here, we justify this idea by showing that we can borrow the calibration analyses of multi-class surrogates and set $\Phi$ to any $(K+1)$-class $\ell_{01}$-calibrated surrogates according to the following conditions:

**Theorem 4.** $L_c^\Phi$ is $\ell_{01c}$-consistent for any $c \in [0, 1]$ if $\Phi$ is an $\ell_{01}$-calibrated surrogate loss. Furthermore, if $\Phi$ is permutation-invariant, i.e., $\Phi(P\boldsymbol{g}) = P\Phi(\boldsymbol{g})$ for all permutation matrices $P$, $\Phi$'s $\ell_{01}$-calibration is also necessary for the $\ell_{01c}$-consistency of $L_c^\Phi$.

The complete proof is shown in Appendix D and here we provide its sketch. The equivalence between CwR and multi-class classification on $\mathcal{D}_c^\circledR$ shown in Theorem 1, 2, and Corollary 1 directly yields the sufficiency of this condition. Though the equivalent classification problem is limited on $\mathcal{D}_c^\circledR$, the permutation-invariance of $\Phi$ and the arbitrariness of $c$ require that the minimizers of the expectation of $\Phi$ should be included in that of $\ell_{01}$ for any potential distributions, which indicates the necessity of $\Phi$'s calibration.

As a result, we can use any $\Phi$ in an off-the-shelf manner, i.e., to the consistency of different $L_c^\Phi$, we only have to check if $\Phi$ is $\ell_{01}$-calibrated, which has been studied thoroughly [7, 54, 47], instead of tedious case-based discussions. It is noticeable that when considering multi-class surrogates that are not permutation-invariant, the calibration of $\Phi$ may not be necessary, which indicates that surrogates that are not calibrated and not permutation-invariant may also be potential options. Nevertheless, a classification-calibrated surrogate $\Phi$ can always be a safe choice given the sufficiency in Theorem 4 and fruitful calibration analyses.

## 5.2 Calibration Result for Generalized Cross Entropy Loss

Given the sufficient condition for $\ell_{01c}$-consistency, we can construct $L_c^\Phi$ with any $\ell_{01}$-calibrated surrogates. However, it has been shown in Charoenphakdee et al. [8] that it can lead to a model that rejects more data than necessary if the cross entropy (CE) loss is used as $\Phi$, which is a popular choice as a surrogate. Another common surrogate is the mean absolute error (MAE). Though it can avoid CPE and only focus on the crucial class with the maximum posterior probability, it usually takes more training epochs before convergence [66], which can be costly in practical use.

Here, we consider the *generalized cross entropy* (GCE) loss [66] that can take the advantages of the CE loss and MAE, which is defined as below:

**Definition 6.** (Generalized cross entropy losses) For any $\gamma \in (0,1]$, the GCE loss is defined as below:

$$\Phi^\gamma(\boldsymbol{g}(\boldsymbol{x}), y) = (1 - S(\boldsymbol{g})_y^\gamma)/\gamma,$$

where $S(\cdot)$ is the softmax-transformation.

It can be seen that the loss formulation is equivalent to MAE if $\gamma = 1$ and it is also reported in Zhang and Sabuncu [66] that the GCE loss can approximate the CE loss if $\gamma \to 0$. Though the GCE loss has proved to be effective in practical use, to the best of our knowledge, its calibration results remain unknown. To justify the combination of GCE loss and $L_c^\Phi$, we give the calibration analysis and further show its analytical solution.

**Theorem 5.** The GCE loss $\Phi^\gamma$ is $\ell_{01}$-calibrated for any $\gamma \in (0,1]$. For the optimal model $\boldsymbol{g}^*$, $S(\boldsymbol{g}^*)_y = \eta_y^{\frac{1}{1-\gamma}} / \sum_{y'=1}^K \eta_{y'}^{\frac{1}{1-\gamma}}$ for all the $\boldsymbol{x} \in \mathcal{X}$ almost surely if $\gamma \in (0,1)$. If $\gamma = 1$, $S(\boldsymbol{g}^*)_{\operatorname{argmax}_y \eta_y} = 1$.

The proof can be found in Appendix E. After verifying the calibration result of the GCE loss, we can combine it with the loss formulation $L_c^\Phi$ and obtain an $\ell_{01c}$-consistent surrogate. We will experimentally demonstrate its effectiveness in the next section.

## 6 Experiments

In this section, we provide the experiment results of CwR with deep models, which are evaluated by the zero-one-$c$ loss following the common practice [44, 8]. We also show the misclassification rate of the accepted data and the ratio of the rejected data. Details of the setup and the experiments for instance-dependent cost can be found in Appendix F and G, respectively.

**Datasets and Models.** In the experiments, we evaluate the proposed methods and baselines on three widely-used benchmarks Fashion-MNIST [60], SVHN [43], CIFAR-10 [31] with cost $c$ selected from $\{0.05, 0.06, 0.07, 0.08, 0.09, 0.10\}$ for Fashion-MNIST and $\{0.05, 0.10, 0.15, 0.20, 0.25, 0.30\}$ the other two. We conduct data augmentation for CIFAR-10 and use the original datasets of Fashion-MNIST and SVHN in the experiments. For Fashion-MNIST, we use a CNN defined in Charoen-phakdee et al. [8], and ResNet-18 and ResNet-34 [26] are used for SVHN and CIFAR-10, respectively.

**Baselines.** We compare our method with state-of-the-art methods in CwR, including confidence-based cross entropy loss (CE) [44], learning to defer (DEFER) [41], and cost-sensitive learning-based method with sigmoid loss (CS) [8], in which DEFER is a special case of our method that use cross entropy loss as $\Phi$. For CE, we also conduct the temperature scaling [25] to alleviate overconfidence. For the proposed method, we use GCE with default parameter $\gamma = 0.7$ as suggested in Zhang and Sabuncu [66] and pairwise-sigmoid (Sigmoid) loss [65] to construct the surrogate $L_c^\Phi$. We implemented all the methods by Pytorch [45], and conducted all the experiments on NVIDIA GeForce 3090 GPUs.

**Experimental Results.** As can be seen from the experimental results reported in Table 2, our proposed method (i.e., either GCE or Sigmoid) significantly outperforms other compared methods in most cases. Obviously, for all the datasets and cost $c$, our GCE method outperforms the baseline DEFER method, which indicates that CwR cannot be simply solved by the methods used for learning to defer. It can be also seen that confidence-based CE is only comparable to the proposed method on FMNIST with a simple CNN. When complex models are used, the effect of overconfidence is inevitable even with the use of temperature scaling, which can be induced from the fact that CE often rejects less data than GCE on SVHN and CIFAR-10. Though CS is comparable to GCE on CIFAR-10 when the rejection cost is high, its performance degrades drastically when the rejection cost decreases, which shows that it is not the best choice in highly error-critical tasks. When ResNet-18 and ResNet-34 are used on SVHN and CIFAR-10 respectively, our GCE method outperforms or is comparable to all the baselines, which shows that GCE is more stable on complex models. Our proposed Sigmoid method performs better than most baselines and is comparable to CE with the use of a simple CNN model, which aligns with the existing observations that pairwise losses are

**Table 2:** The mean and standard error of the zero-one-$c$ losses (**01c**, rescaled to 0-100), rejection ratio (**Rej**), and missclassification rates (**01**) of the accepted data for 5 trails. The best and comparable methods based on the paired t-test at the significance level $5\%$ are highlighted in boldface.

| Method | Cost | CE 01c | CE Rej | CE 01 | CS 01c | CS Rej | CS 01 | DEFER 01c | DEFER Rej | DEFER 01 | GCE 01c | GCE Rej | GCE 01 | Sigmoid 01c | Sigmoid Rej | Sigmoid 01 |
|---|---|---|---|---|---|---|---|---|---|---|---|---|---|---|---|---|
| FMNIST | 0.05 | 2.30 (0.07) | 25.17 (3.17) | 1.39 (0.11) | 2.93 (0.25) | 34.95 (1.94) | 1.81 (0.48) | 3.79 (0.28) | 50.461 (2.51) | 2.58 (0.46) | 3.22 (0.07) | 50.47 (2.49) | 1.39 (0.30) | **2.23** (**0.01**) | 30.98 (0.62) | 0.99 (0.05) |
| | 0.06 | **2.58** (**0.07**) | 22.92 (1.45) | 1.56 (0.09) | 3.37 (0.15) | 33.13 (1.27) | 2.07 (0.28) | 4.63 (0.10) | 56.45 (3.69) | 2.84 (0.42) | 3.78 (0.17) | 50.46 (1.24) | 1.53 (0.30) | **2.62** (**0.08**) | 26.76 (3.37) | 1.37 (0.21) |
| | 0.07 | **2.73** (**0.14**) | 21.17 (2.23) | 1.58 (0.31) | 3.45 (0.17) | 35.77 (2.62) | 1.47 (0.04) | 5.18 (0.47) | 56.46 (6.85) | 2.86 (0.41) | 4.23 (0.21) | 48.05 (5.24) | 1.66 (0.25) | 2.94 (0.07) | 29.87 (0.85) | 1.21 (0.17) |
| | 0.08 | **3.12** (**0.11**) | 20.71 (1.68) | 1.85 (0.07) | 4.13 (0.36) | 33.68 (0.32) | 2.17 (0.52) | 5.86 (0.30) | 54.08 (3.47) | 3.36 (0.29) | 4.50 (2.36) | 45.66 (2.36) | 1.55 (0.25) | **3.14** (**0.17**) | 26.10 (0.23) | 1.43 (0.25) |
| | 0.09 | **3.55** (**0.21**) | 23.64 (1.82) | 1.86 (0.18) | 4.20 (0.21) | 31.90 (1.74) | 1.96 (0.15) | 6.31 (0.40) | 54.62 (4.33) | 3.09 (0.49) | 4.95 (0.06) | 44.05 (1.74) | 1.77 (0.23) | **3.50** (**0.05**) | 23.71 (.28) | 1.79 (0.18) |
| | 0.10 | **3.59** (**0.16**) | 18.32 (1.56) | 2.15 (0.32) | 4.45 (0.20) | 28.96 (0.13) | 2.18 (0.41) | 6.72 (0.07) | 52.69 (0.74) | 3.08 (0.18) | 5.06 (0.23) | 39.01 (4.87) | 1.89 (0.26) | **3.73** (**0.05**) | 23.96 (1.90) | 1.76 (0.20) |
| SVHN | 0.05 | 3.33 (0.14) | 14.37 (0.94) | 3.05 (0.13) | 4.42 (0.13) | 12.81 (0.14) | 4.33 (0.12) | 4.19 (0.29) | 33.05 (1.59) | 3.80 (0.37) | **2.68** (**0.17**) | 19.79 (0.72) | 2.10 (0.24) | **2.70** (**0.14**) | 29.56 (1.16) | 1.73 (0.17) |
| | 0.10 | 4.66 (0.20) | 10.91 (0.57) | 4.01 (0.21) | 4.48 (0.14) | 12.85 (0.42) | 3.67 (0.11) | 5.55 (0.56) | 30.72 (2.64) | 3.58 (0.52) | **4.13** (**0.11**) | 14.83 (0.54) | 3.10 (0.10) | **4.13** (**0.39**) | 19.16 (1.94) | 2.74 (0.43) |
| | 0.15 | 5.40 (0.09) | 8.52 (0.15) | 4.50 (0.07) | 5.14 (0.10) | 13.21 (0.62) | 3.64 (0.19) | 6.37 (0.21) | 21.19 (0.94) | 4.05 (0.25) | **4.66** (**0.06**) | 11.47 (0.41) | 3.31 (0.07) | **4.83** (**0.44**) | 18.38 (1.37) | 2.54 (0.61) |
| | 0.20 | 6.16 (0.13) | 7.74 (0.26) | 4.99 (0.09) | **5.51** (**0.20**) | 12.78 (1.03) | 3.19 (0.24) | 5.99 (0.17) | 12.33 (0.51) | 4.02 (0.16) | **5.44** (**0.04**) | 10.02 (0.25) | 3.82 (0.03) | 6.39 (0.45) | 15.86 (0.72) | 3.82 (0.48) |
| | 0.25 | 7.08 (0.32) | 6.51 (1.06) | 5.83 (0.36) | 6.77 (0.16) | 12.96 (0.97) | 4.06 (0.18) | 6.69 (0.16) | 9.18 (0.35) | 4.33 (0.16) | **5.75** (**0.14**) | 8.64 (0.20) | 3.93 (0.12) | 6.74 (0.13) | 13.79 (0.33) | 3.82 (0.14) |
| | 0.30 | 7.12 (0.16) | 5.31 (0.36) | 5.83 (0.18) | 7.26 (0.33) | 13.21 (1.20) | 3.80 (0.41) | 7.07 (0.31) | 12.35 (2.31) | 4.55 (0.34) | **6.30** (**0.09**) | 8.72 (0.11) | 4.04 (0.09) | 7.69 (0.22) | 10.79 (0.76) | 5.00 (0.13) |
| CIFAR-10 | 0.05 | 4.43 (0.23) | 29.93 (1.85) | 4.18 (0.33) | 6.59 (0.27) | 20.20 (0.51) | 7.00 (0.35) | 4.62 (0.47) | 44.97 (5.24) | 4.30 (0.88) | 3.80 (0.20) | 34.52 (2.77) | 3.16 (0.35) | **3.67** (**0.03**) | 42.69 (8.74) | 2.63 (0.49) |
| | 0.10 | 7.13 (0.11) | 21.13 (0.81) | 6.35 (0.18) | 7.68 (0.32) | 20.31 (0.66) | 7.08 (0.42) | 6.56 (0.26) | 26.21 (1.12) | 5.34 (0.39) | **5.84** (**0.12**) | 25.47 (0.98) | 4.41 (0.15) | **6.11** (**0.13**) | 31.66 (2.17) | 4.30 (0.30) |
| | 0.15 | 9.03 (0.32) | 7.76 (0.39) | 7.74 (0.37) | 8.35 (0.29) | 21.83 (0.92) | 6.49 (0.45) | 8.39 (0.19) | 20.39 (1.59) | 6.69 (0.35) | **7.56** (**0.14**) | 20.43 (0.60) | 5.65 (0.23) | 8.18 (0.10) | 23.39 (0.82) | 6.10 (0.18) |
| | 0.20 | 10.45 (0.29) | 14.53 (0.47) | 8.82 (0.38) | **9.32** (**0.21**) | 21.86 (0.46) | 6.33 (0.33) | 9.65 (0.14) | 17.16 (1.04) | 7.50 (0.11) | **9.09** (**0.14**) | 18.45 (1.93) | 6.62 (0.42) | 9.69 (0.15) | 19.54 (1.55) | 7.20 (0.07) |
| | 0.25 | 11.64 (0.26) | 11.20 (0.30) | 9.96 (0.32) | **10.46** (**0.24**) | 22.02 (0.40) | 6.35 (0.35) | 10.85 (0.08) | 14.22 (0.35) | 8.50 (0.08) | **10.31** (**0.23**) | 15.39 (1.47) | 7.64 (0.38) | 10.96 (0.11) | 14.99 (1.71) | 8.48 (0.40) |
| | 0.30 | 12.20 (0.18) | 10.02 (0.53) | 10.89 (0.15) | **11.43** (**0.23**) | 22.23 (0.81) | 6.13 (0.24) | 11.90 (0.17) | 11.48 (0.75) | 9.55 (0.31) | **11.23** (**0.16**) | 12.52 (0.122) | 8.55 (0.14) | 12.14 (0.12) | 11.08 (0.60) | 9.91 (0.25) |

often effective with simple models [57, 15]. These results show that our method can benefit from the flexibility of the choices of loss functions.

## 7 Conclusion

In this paper, we studied the problem of classification with rejection, which can refrain from making a prediction to avoid critical misclassification. We derived a novel formulation for CwR that can be equipped with arbitrary loss functions while maintaining the theoretical guarantees, making them highly adaptive to the dataset in practical use. First, we showed the equivalence between $K$-class CwR and a $(K+1)$-class classification problem, and proposed an empirical risk minimization formulation to solve this problem with an estimation error bound. Then, we pointed out necessary and sufficient conditions for the learning consistency of the surrogates constructed on our proposed formulation equipped with any classification-calibrated multi-class losses. Finally, experimental results demonstrated the effectiveness of our proposed method.

## Acknowledgement

This research is supported by the National Research Foundation, Singapore under its Industry Alignment Fund – Pre-positioning (IAF-PP) Funding Initiative. Any opinions, findings and conclusions or recommendations expressed in this material are those of the author(s) and do not reflect the views of National Research Foundation, Singapore. Yuzhou Cao was also supported by Ant Group through Ant Research Intern Program. Lei Feng was supported by the National Natural Science Foundation of China (Grant No. 62106028), Chongqing Overseas Chinese Entrepreneurship and Innovation Support Program, and CAAI-Huawei MindSpore Open Fund. MS was supported by JST CREST Grant Number JPMJCR18A2.

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
