# A  Proof of Theorem 1 and Corollary 1

We begin with the proof of Corollary 1 and show that Theorem 1 is its special case.

*Proof.* First of all, we prove that the Bayes optimal solution on $\tilde{p}(\boldsymbol{x}, \tilde{y})$ coincide with the Chow's rule of $p(\boldsymbol{x}, y)$ with cost $c$. According to the optimality condition of multi-class classification, the optimal classifier $f^*(\boldsymbol{x})$ on $\tilde{p}(\boldsymbol{x}, \tilde{y})$ should fulfill the following condition almost surely:

$$f^*(\boldsymbol{x}) = \mathrm{argmax}_{\tilde{y}} \ \tilde{p}(\tilde{y}|\boldsymbol{x}), \tilde{y} \in \{1, \cdots, K, ®\}.$$

According to the definition of $\tilde{p}$, we can further rewrite it as:

$$f^*(\boldsymbol{x}) = \begin{cases} ®, & \max_{\tilde{y} \in \{1, \cdots, K\}} \frac{p(\tilde{y}|\boldsymbol{x})}{2-c(\boldsymbol{x})} \leq \frac{1-c(\boldsymbol{x})}{2-c(\boldsymbol{x})}, \\ \mathrm{argmax}_{\tilde{y} \in \{1, \cdots, K\}} \frac{p(\tilde{y}|\boldsymbol{x})}{2-c(\boldsymbol{x})}, & \text{else}, \end{cases}$$

which coincides with the Chow's rule. Then we have the following conclusions:

$$\bar{R}_{01}(f) = \mathbb{E}_{\tilde{p}(\boldsymbol{x}, \tilde{y})}[(2-c(\boldsymbol{x}))\ell_{01}(f(\boldsymbol{x}), \tilde{y})]$$

$$= \int_{\boldsymbol{x}} \sum_{\tilde{y}=1}^{K} (2-c(\boldsymbol{x}))\ell_{01}(f(\boldsymbol{x}), \tilde{y}) \frac{p(\boldsymbol{x}, \tilde{y})}{2-c(\boldsymbol{x})} d\boldsymbol{x} + \int_{\boldsymbol{x}} (2-c(\boldsymbol{x}))\ell_{01}(f(\boldsymbol{x}), K+1) \frac{p(\boldsymbol{x})}{2-c(\boldsymbol{x})} d\boldsymbol{x}$$

$$= \int_{\boldsymbol{x}} \sum_{\tilde{y}=1}^{K} (2-c(\boldsymbol{x}))\ell_{01}(f(\boldsymbol{x}), \tilde{y}) \frac{p(\boldsymbol{x}, \tilde{y})}{2-c(\boldsymbol{x})} d\boldsymbol{x} + \int_{\boldsymbol{x}} (2-c(\boldsymbol{x}))\ell_{01}(f(\boldsymbol{x}), ®) \frac{(1-c(\boldsymbol{x}))p(\boldsymbol{x})}{2-c(\boldsymbol{x})} d\boldsymbol{x}$$

$$= \int_{\boldsymbol{x}} \sum_{\tilde{y}=1}^{K} \ell_{01}(f(\boldsymbol{x}), \tilde{y})p(\boldsymbol{x}, \tilde{y}) d\boldsymbol{x} + \int_{\boldsymbol{x}} \ell_{01}(f(\boldsymbol{x}), ®)(1-c(\boldsymbol{x}))p(\boldsymbol{x}) d\boldsymbol{x}$$

Suppose $C(f(\boldsymbol{x})) = \sum_{\tilde{y}=1}^{K} \ell_{01}(f(\boldsymbol{x}), \tilde{y})p(\tilde{y}|\boldsymbol{x}) + \ell_{01}(f(\boldsymbol{x}), ®)(1-c(\boldsymbol{x}))$ is the inner risk and $f^*$ is the Chow's rule, we have that

- If $f^*(\boldsymbol{x}) = ®$ and $f(\boldsymbol{x}) \neq f^*(\boldsymbol{x})$:
$$C(f(\boldsymbol{x})) - C(f^*(\boldsymbol{x})) = 1 - c(\boldsymbol{x}) - p(f(\boldsymbol{x})|\boldsymbol{x}).$$

- If $f^*(\boldsymbol{x}) \in \{1, \cdots, K\}$ and $f(\boldsymbol{x}) = ®$:
$$C(f(\boldsymbol{x})) - C(f^*(\boldsymbol{x})) = c(\boldsymbol{x}) - 1 + p(f^*(\boldsymbol{x})|\boldsymbol{x}).$$

- If $f^*(\boldsymbol{x}), f(\boldsymbol{x}) \in \{1, \cdots, K\}$ and $f(\boldsymbol{x}) \neq f^*(\boldsymbol{x})$:
$$C(f(\boldsymbol{x})) - C(f^*(\boldsymbol{x})) = p(f^*(\boldsymbol{x})|\boldsymbol{x})) - p(f(\boldsymbol{x})|\boldsymbol{x}).$$

These conclusion shows that

$$C(f(\boldsymbol{x})) - C(f^*(\boldsymbol{x})) = \mathbb{E}_{p(y|\boldsymbol{x})}[\ell_{01c}(f(\boldsymbol{x}), y)] - \mathbb{E}_{p(y|\boldsymbol{x})}[\ell_{01c}(f^*(\boldsymbol{x}), y)].$$

We can conclude the proof by taking the expectation over $p(\boldsymbol{x})$ on both sides of the equation. $\quad\square$

It can be seen that when $c(\boldsymbol{x})$ is constant, we can divide each side of Corollary 1 to get the proof of Theorem 1.

# B  Proof of Theorem 2

*Proof.*

$$R_{L_c^{\Phi}}(\boldsymbol{g}) = \mathbb{E}_{p(\boldsymbol{x}, y)}[L_c^{\Phi}(\boldsymbol{g}(\boldsymbol{x}), y)]$$
$$= \mathbb{E}_{p(\boldsymbol{x}, y)}[\Phi(\boldsymbol{g}(\boldsymbol{x}), y)] + (1-c)\mathbb{E}_{p(\boldsymbol{x})}[\Phi(\boldsymbol{g}(\boldsymbol{x}), K+1)]$$
$$= (2-c)\tilde{R}_{\Phi}(\boldsymbol{g})$$

$\square$

# C   Proof of Theorem 3

We first give the definition of Rademacher complexity:

**Definition 7.** *(Rademacher complexity [5])* Let $Z_1, \cdots, Z_n$ be n *i.i.d.* random variables drawn from a probability distribution $\mu$ and $\mathcal{F} = \{f : Z \to \mathbb{R}\}$ be a class of measurable functions. Then the expected Rademacher complexity of function class $\mathcal{F}$ is given as follow:

$$\mathfrak{R}_n(\mathcal{F}) = \mathbb{E}_{Z_1, \cdots, Z_n \sim \mu} \mathbb{E}_{\boldsymbol{\sigma}} \left[ \sup_{f \in \mathcal{F}} \frac{1}{n} \sum_{i=1}^{n} \sigma_i f(Z_i) \right], \tag{7}$$

where $\sigma_1, \cdots, \sigma_n$ are the Rademacher variables that take the value from $\{-1, +1\}$ uniformly.

Then we can begin proving Theorem 3.

*Proof.* According to the conditions in Theorem 3, we can learn that $L_c^{\Phi}$ is $(2-c)\rho$-Lipschitz continuous and is bounded by $(2-c)C_{\Phi}$. By applying the McDiarmid's inequality [37], it is routine [40] to show that the following inequalities holds with probability at least $1 - \frac{\delta}{2}$, respectively:

$$\sup_{\boldsymbol{g} \in \mathcal{G}} \left( R_{L_c^{\Phi}(\boldsymbol{g})} - \hat{R}_{L_c^{\Phi}(\boldsymbol{g})} \right) \leq \mathbb{E}_{\boldsymbol{x}_1, \cdots, \boldsymbol{x}_n} \left[ \sup_{\boldsymbol{g} \in \mathcal{G}} \left( R_{L_c^{\Phi}(\boldsymbol{g})} - \hat{R}_{L_c^{\Phi}(\boldsymbol{g})} \right) \right] + (2-c)C_{\Phi} \sqrt{\frac{\log \frac{2}{\delta}}{2n}}$$

$$\sup_{\boldsymbol{g} \in \mathcal{G}} \left( \hat{R}_{L_c^{\Phi}(\boldsymbol{g})} - R_{L_c^{\Phi}(\boldsymbol{g})} \right) \leq \mathbb{E}_{\boldsymbol{x}_1, \cdots, \boldsymbol{x}_n} \left[ \sup_{\boldsymbol{g} \in \mathcal{G}} \left( \hat{R}_{L_c^{\Phi}(\boldsymbol{g})} - R_{L_c^{\Phi}(\boldsymbol{g})} \right) \right] + (2-c)C_{\Phi} \sqrt{\frac{\log \frac{2}{\delta}}{2n}}$$

By applying Talagrand's contraction lemma [36], we can learn that:

$$\mathbb{E}_{\boldsymbol{x}_1, \cdots, \boldsymbol{x}_n} \left[ \sup_{\boldsymbol{g} \in \mathcal{G}} \left( R_{L_c^{\Phi}(\boldsymbol{g})} - \hat{R}_{L_c^{\Phi}(\boldsymbol{g})} \right) \right] \leq \sqrt{2}(2-c)\rho \sum_{y=1}^{K+1} \mathfrak{R}_n(\mathcal{G}_y)$$

and this conclusion also holds for another direction. Plugging this conclusion into the former inequalities and using the union bound, we can learn:

$$\sup_{\boldsymbol{g} \in \mathcal{G}} \left| R_{L_c^{\Phi}(\boldsymbol{g})} - \hat{R}_{L_c^{\Phi}(\boldsymbol{g})} \right| \leq \sqrt{2}(2-c)\rho \sum_{y=1}^{K+1} \mathfrak{R}_n(\mathcal{G}_y) + (2-c)C_{\Phi} \sqrt{\frac{\log \frac{2}{\delta}}{2n}}$$

According to the definition of empirical risk minimization and identifiable condition, we can get the following conclusion, where $\boldsymbol{g}^*$ is the optimal solution among all the measurable functions:

$$
\begin{aligned}
R_{L_c^{\Phi}}(\hat{\boldsymbol{g}}) - \min_{g \in \mathcal{G}} R_{L_C^{\Phi}} &= R_{L_c^{\Phi}}(\hat{\boldsymbol{g}}) - R_{L_c^{\Phi}}^* \\
&= \left( R_{L_c^{\Phi}}(\hat{\boldsymbol{g}}) - \hat{R}_{L_c^{\Phi}}(\hat{\boldsymbol{g}}) \right) + \left( \hat{R}_{L_c^{\Phi}}(\hat{\boldsymbol{g}}) - \hat{R}_{L_c^{\Phi}}(\boldsymbol{g}^*) \right) + \left( \hat{R}_{L_c^{\Phi}}(\boldsymbol{g}^*) - R_{L_c^{\Phi}}^* \right) \\
&\leq \left( R_{L_c^{\Phi}}(\hat{\boldsymbol{g}}) - \hat{R}_{L_c^{\Phi}}(\hat{\boldsymbol{g}}) \right) + \left( \hat{R}_{L_c^{\Phi}}(\boldsymbol{g}^*) - R_{L_c^{\Phi}}^* \right) \\
&\leq 2 \sup_{\boldsymbol{g} \in \mathcal{G}} \left| R_{L_c^{\Phi}}(\boldsymbol{g}) - \hat{R}_{L_c^{\Phi}}(\boldsymbol{g}) \right|
\end{aligned}
$$

which concludes the proof. $\qquad \square$

# D   Proof of Theorem 4

*Proof.* According to Theorem 1, Theorem 2, and Theorem 3 in Ramaswamy and Agarwal [47], we can immediately learn the sufficiency of this condition. We begin the proof of the necessity of the calibration of $\Phi$.

First, we give a useful property for permutation-invariant surrogates. For any $\boldsymbol{p} \in \Delta^{K+1}$, $\mathcal{U}(\boldsymbol{p})$ denotes all the $\boldsymbol{u} \in \mathbb{R}^{K+1} : \boldsymbol{u} \notin \operatorname{argmin}_{\boldsymbol{u}'} \boldsymbol{p}^T \boldsymbol{L}_{01}(\boldsymbol{u}')$ that meet the following condition:

$$\boldsymbol{p}^T \boldsymbol{\Phi}(\boldsymbol{u}) = \inf_{\boldsymbol{u}' \in \mathbb{R}^{K+1}} \boldsymbol{p}^T \boldsymbol{\Phi}(\boldsymbol{u}'),$$

and $\mathcal{P}$ denotes the collection of $\boldsymbol{p} \in \Delta^{K+1}$ that $\mathcal{U}(\boldsymbol{p})$ is non-empty. Given the assumption of permutation-invariance, we can conclude that if $\boldsymbol{p} \in \mathcal{P}$, the set $\mathcal{P}$ contains all of its re-permutation, since given any permutation matrix $P$ and $\boldsymbol{u} \in \mathcal{U}(\boldsymbol{p})$:

$$
\begin{aligned}
(P\boldsymbol{p})^T \Phi(P\boldsymbol{u}) &= \boldsymbol{p}^T P^T P \Phi(\boldsymbol{u}) \\
&= \boldsymbol{p}^T \Phi(\boldsymbol{u}) \\
&= \inf_{\boldsymbol{u} \in \mathbb{R}^{K+1}} \boldsymbol{p}^T \Phi(\boldsymbol{u}) \\
&= \inf_{\boldsymbol{u} \in \mathbb{R}^{K+1}} \boldsymbol{p}^T P^T P \Phi(\boldsymbol{u}) \\
&= \inf_{\boldsymbol{u} \in \mathbb{R}^{K+1}} (P\boldsymbol{p})^T \Phi(P\boldsymbol{u}) \\
&= \inf_{\boldsymbol{u}' \in \mathbb{R}^{K+1}} (P\boldsymbol{p})^T \Phi(\boldsymbol{u}')
\end{aligned}
$$

which indicates $P\boldsymbol{u} \in \mathcal{U}(P\boldsymbol{p})$ and thus $P\boldsymbol{p} \in \mathcal{P}$.

Then we continue to prove the necessity of $\Phi$'s calibration by contradiction. Suppose $\Phi$ is not classification-calibrated and thus $\mathcal{P}$ is non-empty. Denote with $\tilde{\mathcal{P}}$ the collection of all the potential confidence vectors $\{[\tilde{p}(1|\boldsymbol{x}), \cdots, \tilde{p}(K+1|\boldsymbol{x})]\}$ for all the $c \in [0,1]$. We can learn that $L_c^{\Phi}$ cannot be $\ell_{01c}$-consistent if $\mathcal{P} \cap \tilde{\mathcal{P}}$ is non-empty or we can construct a distribution with density $\hat{p}$ whose support only contains a single point $\boldsymbol{x}'$[3] with posterior probability $\boldsymbol{p}' \in \mathcal{P} \cap \tilde{\mathcal{P}}$ for contradiction.

Then we show that $\mathcal{P} \cup \tilde{\mathcal{P}}$ must be non-empty if $\Phi$ is permutation-invariant and not classification-calibrated. For any $p \in \mathcal{P}$, if $p_{K+1} \leq \frac{1}{2}$, we can learn that $\boldsymbol{p} \in \tilde{\mathcal{P}}$ constructed by $\bar{\boldsymbol{p}} \in \Delta^K$ and cost $c$ following the definition of self-augmented distribution:

$$
c = \frac{1 - 2p_{K+1}}{1 - p_{K+1}}, \ \bar{p}_y = (2-c)p_y = \frac{p_y}{1 - p_{K+1}}, \ \forall y \in \{1, \cdots, K\},
$$

and thus $\mathcal{P} \cap \tilde{\mathcal{P}}$ is non-empty. Can we get a non-empty $\mathcal{P}$ that all its elements have a $(K+1)th$ element that is larger than $\frac{1}{2}$? The answer is no: for any $\boldsymbol{p} \in \Delta^{K+1}$, $\min_{\tilde{y} \in \{1, \cdots, K+1\}} \boldsymbol{p}_{\tilde{y}} < 1/2$ and the fact that $\mathcal{P}$ is closed *w.r.t.* the operation of permutation indicates that we can exchange the minimal value in $\boldsymbol{p}$ and its $(K+1)$th element to get a $\boldsymbol{p}' \in \mathcal{P}$ and $\boldsymbol{p}'_{K+1} < 1/2$.

In conclusion, $\mathcal{P} \cup \tilde{\mathcal{P}}$ must be non-empty if $\Phi$ is not classification-calibrated and permutation-invariant, which concludes the proof of necessity. □

# E  Proof of Theorem 5

*Proof.* According to [61], we can directly get the formulation of the optimal solution of GCE. Based on this formulation, we prove the classification-calibration of GCE constructively by giving an regret transfer bound.

First of all, we show that the excess error of GCE loss for any $\boldsymbol{x}$ is a reweighted version of the Tsallis relative entropy [23, 49] in actual. Denote by $S(\boldsymbol{g}^*)_y = \boldsymbol{q}_y^*$, $S(\boldsymbol{g})_y = \boldsymbol{q}_y$ for any $\boldsymbol{g}$, and $p(y|\boldsymbol{x}) = \eta_y$. We substitute $\gamma$ with $r$ in the proof for simplicity:

$$
\begin{aligned}
Ex(\boldsymbol{q}, \boldsymbol{x}) &= \sum_{y=1}^{y} \eta_y \frac{(1 - \boldsymbol{q}_y^r)}{r} - \sum_{y=1}^{y} \eta_y \frac{(1 - \boldsymbol{q}_y^{*r})}{r} \\
&= \frac{\sum_{y=1}^{K} \eta_y (\boldsymbol{q}_y^{*r} - \boldsymbol{q}_y^r)}{r} \\
&= \left( \sum_{y=1}^{K} \eta_y^{\frac{1}{1-r}} \right)^{1-r} \frac{\left(1 - \sum_{y=1}^{K} q_y^{*(1-r)} q_y^r\right)}{r}
\end{aligned}
$$

It can be seen that the second term of the last equation is the Tsallis relative entropy between discrete possibilities $\boldsymbol{q}^*$ and $\boldsymbol{q}$. According to the Corollary 9 of [23] and (4.13) of [49], we can lower bound

---

[3]An equivalent description is that $\hat{p}(\boldsymbol{x}) = \delta(\boldsymbol{x} - \boldsymbol{x}')$, where $\delta(\cdot)$ is the Dirac Delta function.

the excess error with the total variation distance between $\boldsymbol{q}^*$ and $\boldsymbol{q}$ and get a Pinsker's type inequality:

$$Ex(\boldsymbol{q}, \boldsymbol{x}) \geq \left( \sum_{y=1}^{K} \eta_y^{\frac{1}{1-r}} \right)^{1-r} \frac{1-r}{2} \|\boldsymbol{q}^* - \boldsymbol{q}\|_1^2$$

Then we have to connect the *r.h.s.* of the inequality to the excess error w.r.t. 0-1 loss. When $\operatorname{argmax}_y q_y(\boldsymbol{x}) \neq \operatorname{argmax}_y \eta_y$, denote by $\operatorname{argmax}_y q_y(\boldsymbol{x}) = pred$ and $\operatorname{argmax}_y \eta_y = max$:

$$\begin{aligned}
\|\boldsymbol{q}^* - \boldsymbol{q}\|_1 &= \sum_{y=1}^{K} |q_y^* - q_y| \\
&\geq |q_{max}^* - q_{max}| + |q_{pred}^* - q_{pred}| \\
&\geq |q_{max}^* - q_{pred}^* + q_{pred} - q_{max}|
\end{aligned}$$

According to the formulation of the optimal solution of GCE, we can learn that $q_{max}^* \geq q_{pred}^*$. Since $\operatorname{argmax}_y q_y(\boldsymbol{x}) \neq \operatorname{argmax}_y \eta_y$, we can learn that $q_{pred} \geq q_{max}$. Then we can further learn that:

$$\begin{aligned}
\|\boldsymbol{q}^* - \boldsymbol{q}\|_1 &\geq |q_{max}^* - q_{pred}^*| \\
&= \left( \sum_{y=1}^{K} \eta_y^{\frac{1}{1-r}} \right)^{-1} |\eta_{max}^{\frac{1}{1-r}} - \eta_{pred}^{\frac{1}{1-r}}| \\
&= \left( \sum_{y=1}^{K} \eta_y^{\frac{1}{1-r}} \right)^{-1} (\eta_{max}^{\frac{1}{1-r}} - \eta_{pred}^{\frac{1}{1-r}}) \\
&= \left( \sum_{y=1}^{K} \eta_y^{\frac{1}{1-r}} \right)^{-1} (\eta_{max} * \eta_{max}^{\frac{r}{1-r}} - \eta_{pred} * \eta_{pred}^{\frac{r}{1-r}}) \\
&\geq \left( \sum_{y=1}^{K} \eta_y^{\frac{1}{1-r}} \right)^{-1} (\eta_{max} * \eta_{max}^{\frac{r}{1-r}} - \eta_{pred} * \eta_{max}^{\frac{r}{1-r}}) \\
&= \left( \sum_{y=1}^{K} \eta_y^{\frac{1}{1-r}} \right)^{-1} \eta_{max}^{\frac{r}{1-r}} (\eta_{max} - \eta_{pred})
\end{aligned}$$

Then we can learn that:

$$\begin{aligned}
Ex(\boldsymbol{q}, \boldsymbol{x}) &\geq \left( \sum_{y=1}^{K} \eta_y^{\frac{1}{1-r}} \right)^{-1-r} \eta_{max}^{\frac{2r}{1-r}} * \frac{1-r}{2} (\eta_{max} - \eta_{pred})^2 \\
&= \frac{1}{\left( \sum_{y=1}^{K} \eta_y^{\frac{1}{1-r}} \right)^{1+r}} \eta_{max}^{\frac{2r}{1-r}} * \frac{1-r}{2} (\eta_{max} - \eta_{pred})^2 \\
&\geq \frac{1}{\left( \sum_{y=1}^{K} \eta_y^{\frac{1}{1-r}} \right)^{1+r}} \frac{1-r}{2K^{\frac{2r}{1-r}}} (\eta_{max} - \eta_{pred})^2 \qquad (8) \\
&\geq \frac{1-r}{2K^{\frac{2r}{1-r}}} (\eta_{max} - \eta_{pred})^2 \qquad (9)
\end{aligned}$$

The derivation of (8) to (9) is shown in the end of this proof. Then we have the following regret transfer bound:

$$\begin{aligned}
R_{01}(\operatorname*{argmax}_y \boldsymbol{g}_y) - R_{01}^* &\leq \mathbb{E}_{p(\boldsymbol{x})} \left[ \sqrt{C Ex(\boldsymbol{q}, \boldsymbol{x})} \right] \\
&\leq \sqrt{C \mathbb{E}_{p(\boldsymbol{x})} [Ex(\boldsymbol{q}, \boldsymbol{x})]} \quad \text{(Jensen's inequality)} \\
&= \sqrt{C (R_G(\boldsymbol{g}) - R_G^*)}
\end{aligned}$$

where $C = \frac{2K^{\frac{2r}{1-r}}}{1-r}$, $R_G$ is the expected version of GCE loss, and $R_G^*$ and $R_{01}^*$ are the optimal value of the expected version of GCE loss and 0-1 loss, respectively. From this bound, we constructively prove the classification-calibration of GCE loss with $r \in (0, 1)$.

**Proof of (8) to (9):** To complete the step, we only have to show that the term $\dfrac{1}{\left(\sum_{y=1}^K \eta_y^{\frac{1}{1-r}}\right)^{1+r}}$ achieves its minima at any $e_y$, where $e_y = [0, \cdots, 1, \cdots, 0]$ that has the only non-zero value at dimension $y$. Notice that the following conclusion about $p$-norm holds for any real-valued $K$-dimensional vector and $r \in (0, 1)$:

$$\|\boldsymbol{x}\|_{\frac{1}{1-r}} \leq \|x\|_1.$$

Suppose $\boldsymbol{\eta}$ is the vector consists of $\{\eta_y\}_{y=1}^K$. Then:

$$\sum_{y=1}^K \eta_y^{\frac{1}{1-r}} = \|\boldsymbol{\eta}\|_{\frac{1}{1-r}}^{\frac{1}{1-r}}$$

$$\leq \|\boldsymbol{\eta}\|_1^{\frac{1}{1-r}}$$

$$= 1$$

Notice that when $\boldsymbol{\eta} = e_y$ for some $y$, $\sum_{y=1}^K \eta_y^{\frac{1}{1-r}} = 1$, which indicates that $\sum_{y=1}^K \eta_y^{\frac{1}{1-r}}$'s maximum value is 1 when confined in probability simplex $\Delta^K$. Then we can further learn that $\dfrac{1}{\left(\sum_{y=1}^K \eta_y^{\frac{1}{1-r}}\right)^{1+r}} \geq 1$, which concludes the proof. $\square$

It is noticeable that the bound does not hold for $r = 1$, e.g., the case of MAE loss, and the regret transfer bound becomes less compact when $r$ increases. We prove the classification-calibration of MAE loss by showing its regret transfer bound.

**Corollary 2.** Suppose the expected version of MAE loss is $R_M(\boldsymbol{g})$ and its minimal value is $R_M^*$. Then we have:

$$R_{01}(\operatorname*{argmax}_y \boldsymbol{g}_y) - R_{01}^* \leq K(R_M(\boldsymbol{g}) - R_M^*).$$

*Proof.* Given the formulation of the optimal solution $\boldsymbol{q}^*$ of expected MAE loss in Theorem 5, for any $\boldsymbol{x}$, the excess error can be written as:

$$Ex(\boldsymbol{q}, \boldsymbol{x}) = \sum_{y=1}^K \eta_y(1 - q_y) - \sum_{y=1}^K \eta_y(1 - q_y^*)$$

$$= \sum_{y=1}^K \eta_y(q_y^* - q_y)$$

$$= \eta_{max} - \sum_{y=1}^K \eta_y q_y$$

When When $\operatorname{argmax}_y q_y(\boldsymbol{x}) \neq \operatorname{argmax}_y \eta_y$:

$$\eta_{max} - \sum_{y=1}^K \eta_y q_y = \eta_{max} - \eta_{pred} q_{pred} - \sum_{y \neq pred}^K \eta_y q_y$$

$$\geq \eta_{max} - \eta_{pred} q_{pred} - \eta_{max}(1 - q_{pred})$$

$$= q_{pred}(\eta_{max} - \eta_{pred})$$

$$\geq \frac{1}{K}(\eta_{max} - \eta_{pred}),$$

which concludes the proof by taking the expectation on both sides. $\square$

Combine the conclusions above and we can conclude the proof. Though the bound for GCE becomes less tight when $r$ increases, the MAE loss has a better regret transfer bound, which indicates that the regret transfer bound of GCE for $r \in (0, 1)$ may not be good enough. A potential reason is that [23, 49] considered the general case of Tsallis relative entropy while we only need the case that $\boldsymbol{q}$ is a probability distribution. It is promising to further tighten this bound by modifying the conclusions in [23, 49] and limiting $\boldsymbol{q}$ to a $K - 1$-dimensional probability simplex.

# F    Details of the Experiment Setup

## F.1    Detailed Information of Benchmark Datasets

In the experiments, we used 3 widely-used benchmark datasets. Here, we report the sources of these datasets and the way we split them.

- Fashion-MNIST [60]. It is a 10-class dataset of fashion items. Each instance is a 28*28 grayscale image. Source: https://github.com/zalandoresearch/fashion-mnist.
- SVHN [43] It is a 10-class dataset for 10 different digits and each instance is a 32*32*3 colored image in RGB format. Source: http://ufldl.stanford.edu/housenumbers/.
- CIFAR-10 [31]. It is a 10-class dataset for 10 different objects and each instance is a 32*32*3 colored image in RGB format. Source: https://www.cs.toronto.edu/~kriz/cifar.html.

For Fashion-MNIST and SVHN, we trained models on the whole training dataset. For CIFAR-10, we splited 10% of the training dataset as the validation set and conducted random crop and flips for data augmentation. The cost $c$ is less than $0.5$ as suggested in [48] and further decreased on Fashion-MNIST since it is a less difficult dataset.

## F.2    Detailed Information of the Models and Optimization Algorithm

For Fashion-MNIST, we used the model defined in [8] for the experiments. For SVHN and CIFAR-10, ResNet-18 and ResNet-34 is used, respectively. For the cost-sensitive method [8], we use batch normalization [28] at the output layer as suggested in [8] since it fails to work without this modification.

Adam with default momentum was used for optimization in this paper. For Fashion-MNIST, the epoch number, batch size, learning rate, and weight decay are set to 20, 256, 1e-3, and 1e-4. For SVHN, the epoch number, batch size, learning rate, and weight decay are set to 20, 1024, 1e-3, and 1e-4. For CIFAR-10, the epoch number, batch size, learning rate, and weight decay are set to or selected from 200, 1024, {1e-3, 2e-3, 3e-3}, and 1e-4. For Fashion-MNIST and SVHN, we use the model after the 20th epoch for performance evaluation. For CIFAR-10, we report the performance of the model with the best performance on the validation dataset. Temperature scaling is further conducted for CE on CIFAR-10.

# G    Details of Instance-dependent Rejection Cost

In practical applications, it can be beneficial letting the rejection cost $c(\boldsymbol{x})$ vary among different samples. For example, when constructing a system to automatically prescribe for users, a wrong prescription can be fatal for users of advanced ages or with underlying diseases. To prevent such wrong prescriptions, the cost for this type of users can be decreased to encourage rejection. However, it is not suitable encouraging rejection for all the users, which makes the system meaningless. An acceptable choice is to increase the cost for rejection instead for users of low risk.

In this appendix, we expand the Theorem 2 and propose a surrogate for instance dependent cost based on Corollary 1, whose estimation error bound and calibration analysis can be derived almost symmetrically thanks to the equivalence shown in Corollary 3. Then we further evaluate its performance on SVHN dataset.

## G.1 Expansion of Theorem 2

Theorem 2 tells the equivalence between surrogate risk minimization of $L_c^\Phi$ on $p(\boldsymbol{x}, y)$ and surrogate risk minimization of $\Phi$ on $\tilde{p}(\boldsymbol{x}, \tilde{y})$. Here we expand it to the case of instance-dependent cost.

Given the cost function $c(\boldsymbol{x})$ and any function $\Phi(\cdot) : \mathbb{R}^{K+1} \times \{1, \cdots, K+1\} \to \mathbb{R}^+$:

$$L_{c(\boldsymbol{x})}^\Phi(\boldsymbol{u}, y) = (\Phi(\boldsymbol{u}, y) + (1 - c(\boldsymbol{x}))\Phi(\boldsymbol{u}, K+1))/(2 - c(\boldsymbol{x})).$$

Then we have the following conclusion:

**Corollary 3.** For any $\boldsymbol{g} : \mathcal{X} \to \mathbb{R}^{K+1}$ and $R_{L_{c(\boldsymbol{x})}^\Phi}(\boldsymbol{g}) = \mathbb{E}_{p(\boldsymbol{x}, y)}[L_{c(\boldsymbol{x})}^\Phi(\boldsymbol{g}(\boldsymbol{x}), y)]$:

$$R_{L_{c(\boldsymbol{x})}^\Phi}(\boldsymbol{g}) = \tilde{R}_\Phi(\boldsymbol{g})$$

*Proof.*

$$
\begin{aligned}
R_{L_{c(\boldsymbol{x})}^\Phi}(\boldsymbol{g}) &= \mathbb{E}_{p(\boldsymbol{x}, y)}[L_{c(\boldsymbol{x})}^\Phi(\boldsymbol{g}(\boldsymbol{x}), y)] \\
&= \mathbb{E}_{p(\boldsymbol{x}, y)}[(\Phi(\boldsymbol{g}(\boldsymbol{x}), y) + (1 - c(\boldsymbol{x}))\Phi(\boldsymbol{g}(\boldsymbol{x}), K+1))/(2 - c(\boldsymbol{x}))] \\
&= \int_{\boldsymbol{x}} \sum_{y=1}^{K} \Phi(\boldsymbol{g}(\boldsymbol{x}), y) \frac{p(\boldsymbol{x}, y)}{2 - c(\boldsymbol{x})} d\boldsymbol{x} + \int_{\boldsymbol{x}} \frac{(1 - c(\boldsymbol{x}))p(\boldsymbol{x})}{2 - c(\boldsymbol{x})} \Phi(\boldsymbol{g}(\boldsymbol{x}), K+1) d\boldsymbol{x} \\
&= \tilde{R}_\Phi(\boldsymbol{g})
\end{aligned}
$$

$\square$

The derivation of its estimation error bound is similar to that of Theorem 3 by modifying the upper bound and Lipschitz constant, and the necessity and sufficiency of the $\ell_{01}$-calibration of $\Phi$ can also be proved by utilizing the arbitrariness of $\tilde{p}(\boldsymbol{x}, y)$ as in Appendix D.

## G.2 Experiments on SVHN

In this section, we compare our proposed surrogate $L_{c(\boldsymbol{x})}^\Phi$ with CE and DEFER on SVHN. The cost-sensitive learning-based method [8] is not compared since it cannot tackle the case of instance-dependent cost.

In the experiments, we use SVHN [43] to demonstrate the effectiveness of $L_{c(\boldsymbol{x})}^\Phi$. To generate instance-dependent costs, we split 10% of the training dataset and manually corrupt it into to a binary dataset by aggregating the 10 classes into ['0', '2', '3', '5', '6', '8', '9'] and ['1', '4', '7']. We train a binary classifier with on the corrupted dataset with 10 epochs. Then we further use the obtained classifier on training and testing set to split them into 2 parts. For any $\boldsymbol{x}$ that is classified as ['0', '2', '3', '5', '6', '8', '9'], we set $c(\boldsymbol{x}) = c_1$ and $c_2$ otherwise. In the experiments, Adam with default momentum is used with learning rate, batch size and weight decay set to 1e-3, 1024, and 1e-4, respectively. The model used is ResNet-18.

**Table 3:** The mean and standard error of the zero-one-$c$ losses (**01c**, rescaled to 0-100), rejection ratio (**Rej**), and missclassification rates (**01**) of the accepted data for 5 trails. The best and comparable methods based on the paired t-test at the significance level $5\%$ are highlighted in boldface.

| Method | $(c_1, c_2)$ | CE | | | DEFER | | | GCE | | |
|--------|--------------|------|------|------|-------|------|------|------|------|------|
| | | 01c | Rej | 01 | 01c | Rej | 01 | 01c | Rej | 01 |
| SVHN | (0.50, 0.10) | 8.03 (0.16) | 4.46 (0.54) | 7.60 (0.01) | 8.00 (0.30) | 9.20 (0.72) | 5.07 (0.25) | **7.20** (**0.17**) | 6.73 (6.73) | 5.13 (5.13) |
| | (0.45, 0.15) | 7.80 (0.26) | 4.36 (0.31) | 7.03 (0.23) | 9.10 (0.46) | 9.93 (0.41) | **5.07** (0.42) | 6.93 (**0.31**) | 7.00 (0.35) | 4.70 (0.26) |
| | (0.40, 0.20) | 7.70 (0.10) | 4.50 (0.50) | 6.83 (0.25) | 7.80 (0.26) | 11.13 (1.27) | 5.00 (0.44) | **7.03** (**0.21**) | 7.93 (0.55) | 4.80 (0.17) |
| | (0.35, 0.25) | 7.76 (0.12) | 4.90 (0.20) | 6.67 (0.15) | 7.70 (0.26) | 11.93 (0.45) | 4.80 (0.10) | **6.83** (**0.20**) | 8.43 (0.31) | 4.63 (0.15) |

The experimental results are reported in the table above. It can be seen that in the scenario of instance-dependent cost, the prop osed surrogate with GCE loss still outperforms baseline methods, which aligns with the observations in Section 6.

## H   Limitations and Potential Negative Social Impacts

**Limitations:**   This framework is used for multi-class classification with rejection, while there are also other scenarios for learning with rejection, e.g., AUC optimization with rejection [52]. We believe that extensions to CwR with complex evaluation is a promising future direction.

**Potential Negative Social Impacts:**   Though classification with rejection can be useful in risk-critical missions, it can lead to inefficient services once abused, i.e., used in risk-insensitive missions. This is also the potential negative social impact of all the methods for CwR.

## I   Synthetic Experiments

In this section, we evaluate the performance of our proposed method GCE with $q = 0.7$ on a synthetic dataset and compare it with the bayes optimal model. The synthetic dataset has a class number of 3 and its class-conditional probabilities *w.r.t.* different classes are generated by three different 2-dimensional Gaussian distributions whose means and covariance matrices are shown below:

$$\boldsymbol{\mu}_1 = [-0.3,\ 0.3]^\top,\ \boldsymbol{\mu}_2 = [0.3,\ 0.3]^\top,\ \boldsymbol{\mu}_3 = [0.3, -0.3]^\top,\ \boldsymbol{\Sigma}_1 = \boldsymbol{\Sigma}_2 = \boldsymbol{\Sigma}_3 = \begin{bmatrix} 0.1 & 0 \\ 0 & 0.1 \end{bmatrix}.$$

The class-prior probabilities are $1/3$ for each class. We generate 6000 data points for training a linear model with Adam and 30000 data points for testing. Since we have access to the *pdf* $p(\boldsymbol{x})$ and $p(\boldsymbol{x}, y)$, we can calculate the class-posterior probability for each data point in the testing set and approximate the performance of the bayes optimal model efficiently by directly applying the Chow's rule. We report the performances in the following table, and the notations are further described in the caption of this table. It can be seen that the proposed method can well approximate the optimal model even only linear model is considered.

**Table 4:** The mean values of the zero-one-$c$ losses (**01c**, rescaled to 0-100), rejection ratio (**Rej**), and missclassification rates (**01**) of the accepted data for 5 trails. The performance of the Bayes Optimal solution is calculated following the Chow's rule on the test dataset.

| $c$ | GCE | | | Bayes Optimal | | |
|---|---|---|---|---|---|---|
| | 01c | Rej | 01 | 01c | Rej | 01 |
| 0.10 | 8.54 | 75.51 | 4.04 | 8.41 | 73.33 | 4.44 |
| 0.15 | 12.25 | 67.96 | 6.42 | 11.72 | 61.46 | 6.48 |
| 0.20 | 15.31 | 59.87 | 8.31 | 14.57 | 52.60 | 8.53 |
| 0.25 | 17.90 | 46.01 | 11.84 | 16.93 | 44.41 | 10.50 |

## J   Connections with Learning to Defer

There are two concurrent papers about learning to defer [10, 56], and one of them [10] is closely related to this paper that our proposed surrogate formulation for instance-independent cost can be seen as their special case with a prior knowledge of expert prediction's accuracy. The framework in Charusaie et al. [10] allows the use of any classification-calibrated surrogates for consistent learning to defer. They further compared the cases of joint-learning and two-stage learning by analyzing function spaces of classifiers and rejectors with fixed VC-dimension, and they also provided an active-learning extension of their framework. However, due to the stochastic nature of expert prediction, their regret transfer bound relies on an assumption on the calibration function, while our bound can be applied on any calibrated surrogates.

In fact, our proposed surrogate for instance-dependent rejection cost can provide a potential problem reduction for learning to defer if an extra step is added. Firstly, we can see the prediction accuracy

of the expert on a certain instance $\boldsymbol{x}$, $\mathbb{E}_{\mu_{X,Y|X=\boldsymbol{x}}}[\mathbb{I}(M \neq Y)]$, as the instance-dependent cost $c(\boldsymbol{x})$. Given the fact that we only have $(M_i, Y_i)$ that is drawn i.i.d. from distribution $\mu_{M,Y|X}$, we can split some samples out to train a regression model for predicting $c(\boldsymbol{x})$ with consistency guarantee, and then plug the predicted $c(\boldsymbol{x})$ into the loss formulation in Appendix G.1 to convert the problem of learning to defer into classification with rejection. This baseline can also be considered in the future study of learning to defer.