# OpenReview forum: "Generalizing Consistent Multi-Class Classification with Rejection to be Compatible with Arbitrary Losses"
_NeurIPS.cc/2022/Conference — NeurIPS 2022 Accept_

### Official Review · Reviewer_Nc8b · 2022-07-06

**Rating:** 6
**Confidence:** 4
**Soundness:** 2 fair
**Presentation:** 3 good
**Contribution:** 3 good

**Summary:**

This paper focuses on consistent surrogate losses for multi-class classification with rejection, where consistency means that minimizing the surrogate risk also minimizes the desired target risk. The authors first show that $K$-class classification with rejection is equivalent to a $(K+1)$-class classification problem on a modified probability distribution $\tilde{p}$. They then show how surrogate losses for multi-class classification, operating on the original data distribution, can be modified to compute the risk on the modified distribution $\tilde{p}$. It is claimed that ($\ell_{01}$-)calibration of a surrogate for multi-class classification is a necessary and sufficient condition for consistency of the surrogate for classification with rejection. The paper also provides an estimation error bound for empirical risk minimization (ERM) of surrogates, and establishes the $\ell_{01}$-calibration of the generalized cross entropy (GCE) loss. Experiments compare the proposed approach using two different surrogates, GCE and pairwise-sigmoid, to recent works on classification with rejection on three datasets.

**Questions:**

Please respond to my concerns about Theorems 4 and 5 especially. My current score reflects these concerns.

For curiosity:
- Theorem 3: If the identifiability condition were dropped, then could we simply replace $R_{L_c^\Phi}^*$ in (6) with $\min_{g\in\mathcal{G}} R_{L^\Phi_c} (g)$?
- Lines 286-288, "GCE method outperforms the baseline DEFER method, which indicates that CwR cannot be simply solved by the methods used for learning to defer": I am wondering whether it really indicates that, or rather the advantage of using GCE over CE since that is the difference between the two (see lines 278-279).

**Limitations:**

OK

**Strengths And Weaknesses:**

**Strengths:**
+ The equivalence between $K$-class classification with rejection and $(K+1)$-class classification allows a wide range of existing multi-class surrogate losses to be used, where classification calibration in the more usual sense (without rejection) is sufficient for consistency in classification with rejection.
+ The experiments demonstrate the benefit of this flexibility by showing that the proposed approach with GCE and pairwise-sigmoid losses can outperform existing methods that are restricted in their loss function.

**Weaknesses:**

I have several concerns about the proofs.
- Theorem 4, proof of necessity
    - First, should "suppose there are some $\mathbf{u} \in \Delta^{K+1}$" be $\mathbf{p}$ instead? $\mathbf{u}$ is optimized over in the equation below. If yes, then it would be better to change $\mathcal{U}$ to $\mathcal{P}$ as well.
    - I do not see why "any re-permutation of $\mathbf{u}$ ($\mathbf{p}$) also fulfills the equation above." Does this assume that $\Phi$ is permutation-invariant? Please justify further.
    - The statement that "there always exists a distribution $p(y|\mathbf{x})$ and $c$ that $\tilde{p}(\tilde{y}|\mathbf{x}) \in \mathcal{U}$" also needs further justification in my mind. On the surface, $\tilde{p}(K+1 | \mathbf{x})$ can be at most $1/2$ when $c \to 0$, so this seems like a constraint. Perhaps allowing all permutations in $\mathcal{U}$ (contingent on the point above being true) resolves this, but this should be clarified.
- Theorem 5: Two issues between lines 553, 554
    - In taking the expectation to go from the point-wise excess loss to the excess risk, I think the square root should be on the inside of the expectation, at least to begin with. Perhaps there is then an argument (e.g. Jensen's inequality) to move it to the outside.
    - In the exponent of $K$, I see that the $2r/(1-r)$ comes from the $\eta_\max$ term, but where do the $r + r^2$ terms come from?
- Theorem 1
    - Case $f^*(x) \in \{1, \dots, K\}$, $f(x) = R$ (I am using $R$ for reject because I don't know how to typeset the "registered" symbol in OpenReview): I think the right-hand side is missing $-1 + c(\mathbf{x})$.
    - The $y$ in $p(\mathbf{x}, y)$: I think it should be $\tilde{y}$ since this is the dummy variable being summed over.

**Minor comments:**
- A major thing I was looking for upon reading the introduction is the benefit of allowing a full range of surrogate losses. As I wrote above, I think the experiments have shown that benefit with the GCE and pairwise-sigmoid losses. So I think the introduction can state the benefit of this flexibility more strongly, to better motivate the paper, and point to the results in Section 6.
- "class-posterior possibility" and elsewhere: I believe "probability" is the standard term instead of "possibility".
- Line 48, "family of surrogates are the ensembles of arbitrary binary classification losses": This didn't make sense to me, perhaps words are missing.
- Line 84: "competent" --> "confident"?
- Line 116: "remaining" --> "retaining"
- Line 159, "Lemma 1" : I think this should read "Theorem 1 is a special case of Corollary 1." There seem to be several more broken references to Theorem 1 and/or Corollary 1, e.g. lines 173, 237
- Theorem 3: It would be good to clearly define $\hat{\mathbf{g}}$ as the empirical risk minimizer.
- I was a bit confused by the statements before Theorem 4 about "$\ell_{01c}$-consistency of such surrogates still remaining unchecked." It seems that sufficiency follows naturally from the development of the paper and the proof of Theorem 4 does not add anything on sufficiency. Perhaps these statements could be rephrased to say that what remains in question is the necessity of $\ell_{01}$-calibration (assuming that my concerns about Theorem 4 above can be resolved).
- Line 238: $D_c$ --> $D_c^R$ ($R$ for reject)?
- Line 259: $\Phi^\gamma$ --> $\Phi_\gamma$
- Line 292: "classification cost" --> "rejection cost"?

---

> ### Author Response · Authors · 2022-08-02
> **Response to Reviewer Nc8b (1/2)**
>
> Thank you for your time and constructive comments! Based on your wonderful suggestions, we have polished our proofs and claims in the revised manuscript and the supplemental materials.
>
> **Q 1. Concerns about the proof of Theorem 4:**
>
> A 1. Many thanks for your calling our attention to this issue! We gratefully adopt your suggestions and agreed that permutation-invariance is an indispensable assumption for the necessity of $\Phi$’s calibration. Therefore, we modified the proof and content of Theorem 4. Fortunately, the main conclusion that any calibrated surrogate $\Phi$ can be combined with loss formulation (4) to construct $01c$-consistent surrogate still holds and the correction of previous claims about the necessity in Theorem 4 has little effect on the arrangement of the manuscript.
>
> Specifically, we emphasize the assumption that $\Phi$ is permutation-invariant when discussing the necessity of $\Phi$’s calibration while its sufficiency still holds without this assumption. We also remind in Section 5.1 that the necessity of $\Phi$’s calibration only holds when $\Phi$ is permutation-invariant and suggest that multi-class surrogates that are not permutation-invariant and not classification-calibrated may also be potential options for the construction of $\ell_{01c}$-consistent surrogates when combined with our proposed formulation (4).
>
> Based on the revised version proof and theorem, we have the following responses to the concerns:
>
> **Q 1.1. The usage $u\in\Delta^{K+1}$.**
>
> A 1.1. In the previous version of our manuscript, $u$ represents a $K+1$-dimensional scoring vector, and thus $u$ should be in $R^{K+1}$ instead. We have corrected this typo in the revised version.
>
> **Q 1.2. The permutation-invariance of $\Phi$ should be added. Some points of the proof of necessity should be further justified.**
>
> A 1.2. In the revised version, we add the assumption of permutation-invariance to both the descriptions of Theorem 4 and its proof. We also expand the proof to ensure its completeness.
>
>
> **Q 2. Concerns about the proof of Theorem 5 between lines 553 and 554:**
>
> **Q 2.1. The step of moving the square root out of the expectation should be further justified.**
>
> A. 2.1. Thank you for your suggestion! In the revised manuscript, we add more steps to elaborate the derivation of this step from line 567 to line 569.
>
> **Q 2.2. Where do the terms $r+r^{2}$ in the exponent of $K$ come from?**
>
> A. 2.2. Thank you for raising this concern! We overlooked the sign of the term $-(1+r)$ and thus this term is the result of zooming of term $\left(\sum_{y=1}^{K}\eta_{y}^{\ \frac{1}{1-r}}\right)^{1+r}$ before $\eta_{max}$. After the correction of this flaw, we find that this result is even more compact.
> To make the proof more accessible, we refine the derivation and add more steps in line 547. An extra paragraph is also added at the end of the proof of Theorem 5 to show the new result of the zooming of the term $\frac{1}{\left(\sum_{y=1}^{K}\eta_{y}^{\frac{1}{1-r}}\right)^{1+r}}$.
> Based on the new proof, the constant term $C$ in the regret transfer bound is improved to be smaller than the former one.
>
> **Q 3. In the proof of Theorem 1, term c(x)-1 is missing in the case $f^{*}(x)=1,…,K, f(x)=\textregistered$. Besides, the y in $p(x, y)$ should be $\tilde{y}$ since it is a dummy variable.**
>
> A 3. Thank you for your valuable suggestion! We have added the missing term in the proof and substituted $y$ in $p(x, y)$ by $\tilde{y}$ when they show up in the same formula.
>
> **Q 4. The benefits of allowing a full range of surrogate losses should be shown in the introduction to better motivate this paper.**
>
> A 4. Thank you for raising this concern! We add a highlight sentence in line 58 to emphasize that the abundant choices of surrogate losses can cater for the actual demands of different tasks and datasets.
>
> **Q 5. The statements before Theorem 4 seem confusing and should be rephrased to better serve the purpose of this theorem.**
>
> A 5. Thank you for your constructive suggestion! Since the sufficiency of $\Phi$’s calibration is straightforward, the statement "$\ell_{01c}$-consistency of such surrogates still remains unchecked" seems redundant, and thus we remove it in the revised manuscript.
>
> **Q 6. There are some problems with the use of terminologies and notations.**
>
> A 6. Thank you for your valuable suggestion! We have corrected the typos in the revised manuscript and added the formal definition for the empirically optimal solution $\hat{g}$.

---

> > ### Author Response · Authors · 2022-08-02
> > **Response to Reviewer (2/2)**
> >
> > **Q 7. If the identifiability condition were dropped, then could we simply replace $R_{L_{C}^{\Phi}}^{*}$ in (6) with $\min_{g\in\mathcal{G}}R_{L_{C}^{\Phi}}$?**
> >
> > A 7. Yes. In the proof of Theorem 3, we first bound the uniform convergence, which can be used to derive the estimation error bound for $\min_{g\in\mathcal{G}}R_{L_{C}^{\Phi}}$. Then we plug the identifiability condition into this bound (this step is omitted for simplicity) and finally conclude the proof. Therefore, such replacement is viable. To make the proof more coherent, we add the omitted step in the proof of Theorem 3.
> >
> > **Q 8. If the comparisons between the results of GCE and DEFER really indicate that GCE is better than CE when used as $\Phi$ in the proposed framework?**
> >
> > A 8. Thank you for raising this concern! The experimental results of DEFER, Sigmoid, and GCE are aimed at showing the advantages of the flexible choices of surrogates provided by our formulation. The performance of DEFER indicates that using CE as $\Phi$ is not a generic solution and the provision of a full range of surrogates is non-trivial. In other words, though there can be some datasets and tasks in which DEFER outperforms Sigmoid and GCE, it is not always better than other surrogates as shown in the experiments. To be more rigorous, we also change the sentence “CwR cannot be simply solved by the methods used for learning to defer” to “CwR cannot be simply solved by DEFER" since only DEFER is compared in our experiment, which is included by “methods used for learning to defer” though it is the only method for the task to the best of our knowledge.

---

> > > ### Comment · Reviewer_Nc8b · 2022-08-08
> > > **few more comments on the revised proofs**
> > >
> > > Thanks very much for the revisions. I have a few more comments on the revised proofs of Theorem 4 and 5. The first two are more important, while the rest are minor.
> > >
> > > **Proof of Theorem 4**
> > > 1. My main additional request is to show exactly how "for any $\mathbf{p} \in \mathcal{P}$, if $p_{K+1} \leq 1/2$, we can learn that $\mathbf{p} \in \tilde{\mathcal{P}}$" (line 545). That is, how to set $c$ and $p(x, y)$ (different $p$ than $\mathbf{p}$) in Definition 4 to give a $\tilde{\mathbf{p}} = \mathbf{p}$.
> > > 1. Between lines 534 and 535, the phrase "all the $\mathbf{u} \in \mathbb{R}^{K+1}$ that meet the following condition" is still not clear to me because $\mathbf{u}$ is minimized over in the equation below. Do you mean all the $\mathbf{u}$ that minimize the left-hand side (with the additional constraint) and simultaneously minimize the right-hand side (without the constraint)?
> > > 1. Lines 544 and 550: I would reverse "not classification-calibrated and permutation-invariant" to "permutation-invariant and not classification-calibrated" to make clear that the "not" only negates "classification-calibrated".
> > > 1. Two phrases that I don't understand: "optimal minima" (line 243); "distribution whose support a single point" (line 542-543)
> > >
> > > **Proof of Theorem 5:** In the equation after line 572, right-hand side, should the exponent be $1-r$ or $1 / (1-r)$?

---

> > > > ### Author Response · Authors · 2022-08-08
> > > > **Response to Reviewer Nc8b**
> > > >
> > > > Thank you very much for your constructive comments! We have further revised the proofs and the changes are highlighted in purple.
> > > >
> > > > ***Questions about the proof of Theorem 4***
> > > >
> > > > ***Q1.*** The construction of such $\mathbf{p}\in\tilde{P}$ should be exactly shown.
> > > >
> > > > A1. Thank you for raising this concern! We add the exact values of $c$ and $\bar{\mathbf{p}}\in\Delta^{K}$ in the equations before line 545 to provide the detailed information of the construction of $\mathbf{p}$ based on Definition 4.
> > > >
> > > > ***Q2.*** The phrase "all the $\mathbf{u}\in\mathbb{R}^{K+1}$" should be clarified. Do you mean all the $\mathbf{u}$ that minimize the left-hand side (with the additional constraint) and simultaneously minimize the right-hand side (without the constraint)
> > > >
> > > > A2. Thank you for raising this concern! The meaning mentioned above is actually what we want to present. We find that the phrase and the equation under it used as the definition of $\mathcal{U}(\mathbf{p})$ are quite ambiguous. To eliminate the ambiguity, we have corrected the phrase and equation after line 535.
> > > >
> > > > ***Q3.*** "Not classification-calibrated and permutation-invariant" should be changed to "permutation-invariant and not classification-calibrated".
> > > >
> > > > A3. Thank you for your valuable suggestion! We have revised this point to avoid multiple interpretations.
> > > >
> > > > ***Q4.*** The phrases "optimal minima" and "distribution whose support a single point" are obscure.
> > > >
> > > > A4. Thank you for raising this concern! The phrase "optimal minima" is substituted by a more detailed description.  The phrase "distribution whose support a single point" is changed to "distribution with density $\hat{p}(\mathbf{x})$ whose support only contains a single point" and an equivalent description is also given in the footnote with the help of Dirac Delta function.
> > > >
> > > > ***Question about the proof of Theorem 5***
> > > >
> > > > ***Q1.*** The exponents in the right-hand side of the equation after line 572 should be $\frac{1}{1-r}$.
> > > >
> > > > A1. Thank you for your valuable suggestion! We have corrected the typos in the equations after line 572.

---

> > > > > ### Comment · Reviewer_Nc8b · 2022-08-09
> > > > > **Thanks for the additional details**
> > > > >
> > > > > Thanks again. My last comment concerns the notation $\mathbf{u} \in \mathbb{R}^{K+1} / \arg\min_{\mathbf{u'}} \mathbf{p}^T \mathbf{L}_{01}(\mathbf{u'})$, which implies division. I believe the authors mean "and not in", $\mathbf{u} \in \mathbb{R}^{K+1}: \mathbf{u} \notin \arg\min$... I trust the authors will correct this.
> > > > >
> > > > > I am increasing my score to 6. This reflects the overlooking of permutation invariance and other corrections needed to the proofs of Theorems 4 and 5 in the initial submission.

---

> > > > > > ### Author Response · Authors · 2022-08-09
> > > > > > **Thank you for spending a lot of time and efforts on our paper**
> > > > > >
> > > > > > We have corrected this point and updated it in the supplementary material. We really appreciate your efforts, and thank you again for making this paper more accurate!

---

### Official Review · Reviewer_RQPg · 2022-07-09

**Rating:** 7
**Confidence:** 2
**Soundness:** 3 good
**Presentation:** 2 fair
**Contribution:** 3 good

**Summary:**

This work proposes a ERM learning algorithm for Classification with Rejection(CwR) problems. The work reveals the connection between CwR and traditional multi-class classification problem and avoids leveraging the less tractable self- augmented distribution in the proposed model.  Both the learning bound and consistency of the algorithm are provided. For consistency, this work provides the necessary and sufficient conditions when a surrogate and calibrated loss is adopted.

**Questions:**

In line 162 it says that the minimization of \bar{R}_{01} can be seen as ordinary classification risk minimization with a different marginal density p'(x) which does not affect the calibration since class posterior possibilities remain unchanged. I am a bit confused here, I thought the minimization of \bar{R}_{01}  can be seen as ordinary classification risk minimization only if there is no distribution shift between the two problems. But with p(x) changing, could we still treat them as the same?


**Ethics Review Area:**

["I don’t know"]

**Strengths And Weaknesses:**

Weakness:
1. There are some minor mistakes in the manuscript. For example, in line 159 ‘It is obvious that Lemma1 is a special case of Lemma 1….’’ There is no Lemma 1 in the paper and I assume the author means that corollary 1 is a special case of Theorem 1.
In line173 ’...indicates the minimization of R01c(\phi g) according to Lemmas 1 and 1.
In line 222 ‘According to Lemma1…’’ again, there is no lemma 1 in the paper.
I assume all the lemma1 in the paper refer to corollary 1 but even corollary 1 itself is problematic. Its main body states that ‘the following inequalities hold..’ but all it got is single equality.
Those mistakes (and a few more)  make me confused from time to time during the reading and make the paper a bit harder to follow.
2. I feel some of the important arguments need to be further elaborated. For example, in line 172 ‘with a properly chosen surrogate \phi the minimization of \tilde{R_\phi}(g) can lead to that of \tilde{R}_{01}(\phi(g))’. Although some related works are cited, it would be more helpful for the author to briefly explain how the commonly adopted principles for choosing the proper surrogates and the rationale behind them.


Strength
1. This paper presents an effective ERM algorithm for CwR problems. The learning bound and consistency of the algorithm are both provided.
2. This paper offers an interesting view to connect multi-class classification and CwR, which is able to leverage calibrated surrogate loss to avoid directly addressing the c-self-augmented distribution during the learning process.

---

> ### Author Response · Authors · 2022-08-02
> **Response to Reviewer RQPg**
>
> Thank you for your constructive comments!
>
> **Q 1. There are some minor mistakes in the manuscript that can confuse the readers and make this paper hard to follow.**
>
> A 1. Thank you for raising this concern. We have corrected the misnomers in Corollary 1 and the use of “Lemma” in the revised manuscript, and the relationship between Theorem 1 and Corollary 1 is stated in line 161.
>
> **Q 2. The detailed principles of the selection of surrogate losses mentioned in line 172 and the rationale behind them should be elaborated.**
>
> A 2. Thank you for your valuable suggestion! We have substituted the phrase “properly chosen” with “classification-calibrated” and given its sufficiency for $\ell_{01}$-consistency.
>
> **Q 3. The claimed equivalence between the minimization of $\bar{R}_{01}$ and the minimization of ordinary classification risk seems confusing since the marginal density $p’(x)$ is changed.**
>
> A 3. Thank you for raising this concern! In fact, the phrase ‘seen as’ here is a result of the following equation, which describes the fact that the reweighted classification task always shares the same solution with a classification task that has a shifted marginal density:
>
> $E_{p(x,\ y)}[\bar{\ell_{01}}(f(x),\ y)]=CE_{p’(x,\ y)}[\ell_{01}(f(x),\ y)],$
>
> where $p’(y|x)=p(y|x)$, $p’(x)=p(x)(2-c(x))/C $, and $C=\int_{x’}p(x’)(2-c(x))dx’$. Combing with this equation, we can conclude that there is no need to conduct calibration analyses for this reweighted classification task since calibration is a point-wise concept and $p’(y|x)=p(y|x)$. Therefore, the phrase ‘seen as’ here holds in the sense that the two tasks share the same solution and the same calibration results.

---

> > ### Comment · Reviewer_RQPg · 2022-08-09
> > **Thank you for the response**
> >
> > Thank you for your response. I think it addresses my concerns. I would like to keep my rating as 'accept'.

---

> ### Comment · Area_Chair_X6n2 · 2022-08-08
> **please acknowledge the authors' response**
>
> Please acknowledge the authors' response.

---

### Official Review · Reviewer_6EcA · 2022-07-10

**Rating:** 7
**Confidence:** 3
**Soundness:** 3 good
**Presentation:** 3 good
**Contribution:** 4 excellent

**Summary:**

The paper considers the consistency of surrogate losses in classification with rejection (CwR) problem.

The paper first construct an augmented classification without rejection problem, which is equivalent to the CwR problem. Then,  building upon the classical consistency analyses, the paper defines a family of l_{01c} consistent losses in CwR based on arbitrary l_{01} calibrated loss. Further, it proves that such a relation is a necessary and sufficient condition for l_{01c} consistency.

Although most of the existing losses are proven consistent, the paper found that the generalized cross-entropy loss has not been investigated before and gives the first proof.

The GCE induced loss achieves a good performance in practice.

**Questions:**

I do not have too much to complain about.

**Limitations:**

This is a theoretical work and I did not see a direct negative societal impact.

**Strengths And Weaknesses:**

Strengths:
1. The main contribution of the paper is to show that the CwR problem can be reduced to an augmented classification problem without rejection. The theory, as well as the methodology, are insightful.

2. The theory inspires new surrogate losses in CwR that works well in real data.

Weakness:
1. A very minor concern is that maybe the authors can validate the theory on some synthetic data where everything (e.g, all risks) can be calculated.

---

> ### Author Response · Authors · 2022-08-02
> **Response to Reviewer 6EcA**
>
> Thank you for your valuable comments!
>
> **Q1. Analysis of synthetic datasets can be made to analytically demonstrate the proposed method.**
>
> A 1. Thank you for your valuable suggestion! We have added an experiment in Appendix I on a 3-class dataset generated from 3 different gaussian distributions with a linear model. The 01c risk, rejection ratio, the classification risk of our model, and the corresponding statistics of the Bayes optimal model are also compared in detail.

---

> > ### Comment · Reviewer_6EcA · 2022-08-09
> > **Thanks for the feedback**
> >
> > Thanks for the feedback. I'm glad to see the analysis of synthetic datasets. Overall, I still think this paper should be accepted and I keep my score.

---

> ### Comment · Area_Chair_X6n2 · 2022-08-08
> **please acknowledge the authors' response**
>
> Please acknowledge the authors' response.

---

### Official Review · Reviewer_bLUw · 2022-07-11

**Rating:** 7
**Confidence:** 4
**Soundness:** 4 excellent
**Presentation:** 3 good
**Contribution:** 3 good

**Summary:**

This paper studies classification with rejection (CwR) in multiclass problems. The proposed method can utilize any classification-calibrated surrogate losses while maintaining theoretical guarantees (consistency). Specifically, the work establishes the equivalence between K-class CwR problem and a (K+1)-class classification problem by showing the two classification risks are equal. Then, using any multiclass surrogate loss as a building block, a formulation of surrogates is proposed to realize the surrogate risk of the (K+1)-class classification problem, with only the original K-class training data needed. An estimation error bound is derived for its ERM. It is then shown that a necessary and sufficient condition for the proposed surrogate to be consistent is that the underlying multiclass surrogate loss is classification-calibrated. Finally, the calibration analysis of the generalized cross entropy (GCE) loss is provided.

**Questions:**

- In Section 6, it is mentioned that DEFER is a special case of the proposed method with cross entropy loss as $\Phi$. However, cross entropy loss is also classification-calibrated. Then why the performance of DEFER is not that good?
- For FMNIST, CE and Sigmoid outperformed GCE. Any ideas why that's the case?

**Limitations:**

Yes, in the appendix.

**Strengths And Weaknesses:**

This paper is a solid piece of work. It is also well-written and well-presented.

**Originality:**
The proposed method can utilize any classification-calibrated surrogate losses while maintaining consistency guarantees, by showing the equivalence between K-class CwR problem and a (K+1)-class classification problem. It is novel, simple and effective. Related works have been properly cited and discussed.

**Quality:**
The work is sound. Claims are well supported by both the theory and experimental results. Weaknesses and limitations have been discussed.

**Clarity:**
The paper is mostly well-written and well-organized. Some minor problems I found are listed below. The authors are encouraged to proofread more carefully.
- "Lemma X" are referred to several times, while the paper only has "Theorem" and "Corollary" instead of "Lemma". Line 159. Line 173. Line 222. Line 237.
- Line 173: Should be $R_{01c} (c \circ \mathbf{g})$.
- Definition 6 and Theorem 5: inconsistent notations $\Phi_r$ and $\Phi^r$.
- Section 5.2: the flow can be improved. Before Theorem 5, "..., to the best of our knowledge, its calibration results remain unknown, and thus it is unsafe ...". Then the calibration of GCE is presented in Theorem 5. The flow of this part seems a bit abrupt.
- In Table 2, I think 01c, Rej, and 01 correspond to the first paragraph of Section 6, but it would be more clear to include their meanings in the caption of the table.

**Significance:**
The paper is a valuable piece of work in the field classification with rejection. The proposed method is simple, general, and effective. It can also be very useful for practitioners.

---

> ### Author Response · Authors · 2022-08-02
> **Response to Reviewer bLUw**
>
> Thank you for your insightful suggestions!
>
> **Q 1. There are some problems w.r.t. the clarity of this paper that should be further proofread.**
>
> A 1. Thank you for raising this concern. We have rechecked our manuscript and corrected clarity problems in the revised manuscript.
>
>  **Q 2. Why the performance of DEFER is not that good though its surrogate loss is induced from the cross-entropy loss that is also classification-calibrated?**
>
> A 2. This observation also appears in the previous work [1]. A potential explanation is that cross-entropy loss focuses more on the task of class-posterior probability estimation while Sigmoid and GCE are more targeted at the discriminative classification problem. Moreover, DEFER rejects more data than necessary when the rejection cost is small, which may be caused by DEFER’s attempt to fit the posterior probability $(1-c)/(2-c)$ of the pseudo-class @, which hinders the further refinement of its result.
>
> [1]. Nontawat Charoenphakdee, Zhenghang Cui, Yivan Zhang, and Masashi Sugiyama. Classification with rejection based on cost-sensitive classification. ICML 2021
>
>
> **Q 3. Why CE and Sigmoid outperform GCE on FMNIST?**
>
> A 3. The less challenging nature of FMNIST can underlie the cause of CE’s better performance. Compared with GCE, CE often suffers from a rejection ratio that is lower than necessary caused by over-confidence, which can be seen in the experiments on SVHN and CIFAR-10. However, as a relatively simple dataset, the optimal rejection ratio of FMNIST is intrinsically low and thus CE’s low rejection ratio has little effect on its performance when applied on FMNIST.
>
> Model complexity can be the reason for the better performance of Sigmoid on FMNIST. Due to its less steep curvature, Sigmoid can suffer from inefficient optimization when trained with complex models. However, since we use a simple CNN on FMNIST, the training of this model with Sigmoid loss is less challenging. Therefore, Sigmoid achieves better performance on FMNIST. It can be observed from the results of SVHN and CIFAR-10 that with the increasing model complexity, the performance of Sigmoid degrades gradually and is less favorable compared with GCE.

---

> ### Comment · Area_Chair_X6n2 · 2022-08-08
> **please acknowledge the authors' response**
>
> Please acknowledge the authors' response.

---

### Author Response · Authors · 2022-08-02
**General response**

Respected Reviewers,

Thank you so much for your valuable feedback! We have highlighted the major changes of the revised manuscript in blue. We are pleased to answer any additional questions that you might have during the discussion period!

---

### Meta-Review · Area_Chair_X6n2 · 2022-08-23

**Recommendation:** Accept
**Confidence:** Certain

**Metareview:**

The consensus view was that this was solid and practically relevant theoretical research.


**Award:**

No

---

### Decision · Program_Chairs · 2022-09-14

Accept